



# Land use alters dominant water sources and flow paths in tropical montane catchments in East Africa

Suzanne R. Jacobs[1,2,3,4*], Edison Timbe[5], Björn Weeser[3,6], Mariana C. Rufino[2,7], Klaus Butterbach-Bahl[1,4], Lutz Breuer[3,6]

[1]Karlsruhe Institute of Technology – Institute of Meteorology and Climate Research, Atmospheric Environmental Research (KIT/IMK-IFU), Kreuzeckbahnstr. 19, 82467 Garmisch-Partenkirchen, Germany
[2]Centre for International Forestry Research (CIFOR), c/o World Agroforestry Centre, United Nations Avenue, Gigiri, P.O. Box 30677 – 00100 Nairobi, Kenya
[3]Institute for Landscape Ecology and Resources Management (ILR), Justus Liebig University, Heinrich-Buff-Ring 26, 35392 Giessen, Germany
[4]Mazingira Centre, International Livestock Research Institute (ILRI), P.O. Box 30709 – 00100 Nairobi, Kenya
[5]Faculty of Agricultural Sciences, University of Cuenca, Av. Doce de Octubre, Cuenca, Ecuador
[6]Centre for International Development and Environmental Research (ZEU), Justus Liebig University, Senckenbergstr. 3, 35390 Giessen, Germany
[7]Lancaster Environment Centre, Lancaster University, Lancaster LA1 4YQ, United Kingdom
*Current address: Centre for International Development and Environmental Research (ZEU), Justus Liebig University, Senckenbergstr. 3, 35390 Giessen, Germany

*Correspondence to*: Suzanne R. Jacobs (suzanne.r.jacobs@zeu.uni-giessen.de)

**Abstract.** Conversion of natural forest to other land uses could lead to significant changes in catchment hydrology, but the nature of these changes has been insufficiently investigated in tropical montane catchments, especially in Africa. To address this knowledge gap, we identified stream water sources and flow paths in three tropical montane sub-catchments (27–36 km²) with different land use (natural forest, smallholder agriculture and commercial tea plantations) within a 1 021 km² catchment in the Mau Forest Complex, Kenya. Weekly samples were collected from stream water, precipitation and soil water for 75 weeks and analysed for stable water isotopes ($\delta^2$H and $\delta^{18}$O) for mean transit time estimation, whereas trace element samples from stream water and potential end members were collected over a period of 55 weeks for end member mixing analysis. Stream water mean transit time was similar (~4 years) in the three sub-catchments, and ranged from 3.2–3.3 weeks in forest soils and 4.5–7.9 weeks in pasture soils at 15 cm depth to 10.4–10.8 weeks in pasture soils at 50 cm depth. The contribution of springs and wetlands to stream discharge increased from 18, 1 and 48 % during low flow to 22, 51 and 65 % during high flow in the natural forest, smallholder agriculture and tea plantation sub-catchments, respectively. The dominant stream water source in the tea plantation sub-catchment was spring water (56 %), while precipitation was dominant in the smallholder agriculture (59 %) and natural forest (45 %) sub-catchments. These results confirm that catchment hydrology is strongly influenced by land use, which could have serious consequences for water-related ecosystem services, such as provision of clean water.





# 1    Introduction

Tropical montane forests are under high anthropogenic pressure through deforestation. Evidence from tropical montane regions in Central and South America shows that conversion of montane forests to pastures increases the contribution of surface runoff to streamflow, caused by changes in flow paths and stream water sources (Ataroff and Rada, 2000; Germer et al., 2010; Muñoz-Villers and McDonnell, 2013). This could affect the timing and quantity of water supply through reduced infiltration of precipitation and increased occurrence of flood events, and could decrease water quality as a result of soil erosion. In Africa, where much of the population relies on surface water as main water source, understanding the effect of land use change on water supply and quality is crucial to manage resources sustainably. However, the hydrological functioning of tropical catchments is generally less well understood than that of temperate catchments. This is specifically true for tropical montane forest catchments, as those have received less attention in hydrological research compared to the tropical lowlands. Nevertheless, tropical montane forests are known for their high biodiversity (Burgess et al., 2007; Martínez et al., 2009) and provision several other important ecosystem services, including carbon storage (Spracklen and Righelato, 2014) and water supply (Célleri and Feyen, 2009; Martínez et al., 2009).

Stable water isotopes ($^2$H, $^{18}$O) provide a useful tool to study the movement of water through a catchment (McGuire and McDonnell, 2007). Stable isotopes, usually expressed as the ratio of heavy to light isotopes (e.g. $^2$H to $^1$H) relative to a known standard (e.g. VSMOW, the Vienna Standard Mean Ocean Water), are useful tracers in hydrology, since these enter the environment naturally through precipitation. The isotopic composition of water only changes due to mixing with other water sources and fractionation by evaporation and condensation. Due to decreasing costs of analysis, stable isotope-based methods are used more frequently worldwide to trace water through catchments and to identify the origin and flow paths of water inputs to streams. Most case studies in tropical montane areas are from Latin America (e.g. Correa et al., 2017; Crespo et al., 2012; Mosquera et al., 2016b; Roa-García and Weiler, 2010; Timbe et al., 2014; Windhorst et al., 2014), whereas no data is available from African tropical montane catchments.

Mean transit time (MTT), i.e. the time required for rainfall to reach the stream, is a good indicator to assess flow paths, water storage capacity and mixing at the catchment scale (Asano and Uchida, 2012). Transit time also has implications for water quality, since the contact time between water and the soil will affect the chemical composition of the water that finally enters the stream through biogeochemical processes (McGuire and McDonnell, 2006). MTT can be influenced by catchment soil cover (Capell et al., 2012; Rodgers et al., 2005; Soulsby et al., 2006), soil depth, hydraulic conductivity and topographic parameters, such as slope (Heidbüchel et al., 2013; Mosquera et al., 2016b; Muñoz-Villers et al., 2016) or a combination of these factors (Hrachowitz et al., 2009). Changes in vegetation cover and especially soil hydraulic properties as consequence of changes in land management can also modify MTT.

Other naturally occurring tracers, such as the elements Ca, Mg, K, Na and Fe, can also be used to study water flow through a catchment, for example through end member mixing analysis (EMMA). In EMMA, stream water is assumed to be a mixture of different 'end members' or water sources, such as precipitation, throughfall, groundwater and soil water (Christophersen et



al., 1990). A quantification of the contribution of different end members in a catchment provides relevant insight into dominant flow paths and stream water sources (Barthold et al., 2010; Burns et al., 2001; Correa et al., 2017; Crespo et al., 2012; Soulsby et al., 2003) or water provenance (Fröhlich et al., 2008a, 2008b). Application of EMMA in the south-western Amazon revealed, for example, a higher contribution of surface runoff in catchments converted from forest to pasture (Chaves et al., 2008; Neill

et al., 2011). Increased surface runoff could result in higher soil erosion and changes in flow paths that generally affect transport of solutes and contaminants to streams, potentially resulting in decreased water quality.

The Mau Forest Complex in western Kenya is the largest tropical montane rainforest in the country and considered a major 'water tower', supplying fresh water to approximately 5 million people living downstream (Kenya Water Towers Agency, 2015). However, conversion of forest to agricultural land resulted in a 25 % forest loss in the past decades (Kinyanjui, 2011).

This has supposedly led to changes in flow regime (Baldyga et al., 2004; Mango et al., 2011; Mwangi et al., 2016) and increased surface runoff (Baker and Miller, 2013). These observations strongly suggest changes in dominant flow paths as a consequence of land use change, but no scientific evidence is available to confirm this. In this study, we combined MTT analysis and EMMA (Crespo et al., 2012; Katsuyama et al., 2009) to assess the effect of land use on spatial and temporal dynamics of water sources and flow paths in catchments with contrasting land use (i.e. natural forest, smallholder agriculture and commercial tea and tree

plantations) in the Mau Forest Complex. This knowledge is essential in the tropics, where population growth puts significant pressure on forests and water resources, but where little is known about the consequences of deforestation. Previous studies in the South-West Mau block of the Mau Forest Complex observed reduced infiltration rates in agricultural compared to forested land use types (Owuor et al., 2018). Furthermore, analysis of nitrate concentration–discharge relationships of rainfall events suggested more surface runoff in catchments dominated by smallholder agriculture or commercial tea and tree plantations than

in a montane forest catchment (Jacobs et al., in review). Based on these results, we hypothesised that (a) the natural forest catchment has a longer MTT than the tea and tree plantation catchment and the smallholder catchment, and (b) precipitation contributes more to streamflow in the smallholder catchment, followed by the tea and tree plantation catchment and forest catchment. Furthermore, we expected that (c) the contribution of different end members varies throughout the year due to seasonality in rainfall.

## 2   Methods

### 2.1   Study area

This study was conducted in the South-West Mau block of the Mau Forest Complex, western Kenya (Fig. 1, Table 1). Three sub-catchments (27–36 km²) were characterised by different land use types: natural forest (NF), smallholder agriculture (SHA) and commercial tea and tree plantations (TTP). These were nested in a 1 021 km² large catchment, referred to as the main

catchment (OUT), which is characterized by a mixture of these three land use types. The natural forest is classified as Afromontane mixed forest, with species including *Podocarpus milanjianus*, *Juniperus procera* and *Olea hochstetteri* (Kinyanjui, 2011; Krhoda, 1988). The vegetation changes into bamboo forest (*Arundinaria alpina*) above 2 300 m elevation.





The north-western side of the forest, bordering smallholder agriculture, is degraded through encroachment of farms, livestock grazing, charcoal burning and logging (Bewernick, 2016). The smallholder agriculture area is characterised by small farms of less than 2 ha, where beans, maize, cabbage and potatoes are grown interspersed with grazing fields for livestock and small woodlots of *Eucalyptus*, *Pinus* and *Cypressus*. The riparian zones are severely degraded by vegetation clearance for grazing or cultivation and access to the river by humans and livestock. Commercial tea plantations, covering approximately 20 000 ha, are found at lower elevation (1 700–2 200 m) closer to Kericho town (0°22'08" S, 35°17'10" E) and consist of a mosaic of tea fields and *Eucalyptus* plantations, the latter mainly being used for tea processing. Riparian forests of up to 30 m width are well-maintained and contain native tree species, such as *Macaranga kilimandscharica*, *Polyscias kikuyuensis*, *Olea hochstetteri* and *Casearia battiscombei* (Ekirapa and Shitakha, 1996). A more detailed description of land use in the study area can be found in Jacobs et al. (2017).

The geology in OUT originates from the early Miocene, with the lower part, encompassing NF and TTP, dominated by phonolites and the upper part, covering SHA, by phonolitic nephelinites with a variety of Tertiary tuffs (Binge, 1962; Jennings, 1971). The soils are deep and well-drained, classified as humic Nitisols (ISRIC, 2007; Krhoda, 1988). The area has a bi-modal rainfall pattern with highest rainfall between April and July (long rains) and October and December (short rains). January to March are the driest months. Long-term annual precipitation at 2 100 m elevation is 1 988±328 mm yr$^{-1}$ (Jacobs et al., 2017).

## 2.2 Hydroclimatic instrumentation

Hydroclimatic data has been measured in the study area since October 2014 at a 10 minute interval (Jacobs et al., in review). Water level data was recorded at the outlet of each catchment with a radar based sensor (VEGAPULS WL61, VEGA Grieshaber KG, Schiltach, Germany). Discharge was estimated from this data using a site-specific second order polynomial rating curve (Jacobs et al., in review). Nine tipping bucket rain gauges (Theodor Friedrichs, Schenefeld, Germany and ECRN-100 high resolution rain gauge, Decagon Devices, Pullman WA, USA) were installed in the study area across an elevation gradient of 1 717 to 2 602 m (Fig. 1). Each tipping bucket recorded cumulative precipitation (resolution of 0.2 mm per tip) per 10 minutes. Precipitation in each catchment was calculated using Thiessen polygons.

## 2.3 Sampling and laboratory analysis

Each catchment had one site with a precipitation and throughfall sampler, constructed of a 1 litre glass bottle covered with aluminium foil and a funnel of 12.5 cm diameter with a table tennis ball to reduce sample fractionation due to evaporation (Windhorst et al., 2013). The throughfall sampler was placed inside the forest, underneath maize or sugar cane (depending on growing season) and underneath tea bushes in NF, SHA and TTP, respectively. The main catchment only had a precipitation sampler. Additionally, a passive capillary wick sampler was installed in each catchment to collect soil water (Brown et al., 1989). Three PE plates of 30 by 30 cm were inserted horizontally at 15, 30 and 50 cm depth in the soil with as little disturbance of the soil above and around the plate as possible. A glass fibre wick was unravelled and draped on top of each plate to maximize surface area. The remaining wick length was led through a hosepipe to a 1 litre glass bottle, which was placed at 1





to 1.5 m depth in the soil. The installation of all samplers was carried out in September 2015 and stable isotope samples were collected from 15 October 2015 to 17 March 2017. Stream water samples were taken at the outlet of all catchments on a weekly basis. The samples were filtered with 0.45 µm polypropylene filters (Whatman Puradisc 25 syringe filter, GE Healthcare, Little Chalfont, UK or KX syringe filter, Kinesis Ltd., St. Neods, UK) and stored in 2 ml glass vials with screw cap. Weekly

integrated stable isotope samples were collected from the wick, precipitation and throughfall samplers. Water samples were analysed for isotopic composition in the laboratory of Justus Liebig University Giessen, Germany, with cavity ring-down spectroscopy (Picarro, Santa Clara CA, USA). Precipitation water samples from all four sites were used to calculate the local meteoric water line (LMWL) with a linear regression model and the 95 % confidence interval was estimated for the slope and intercept. Only samples with a sampling volume of more than 100 ml were included to avoid the effect of evaporative

enrichment of small sample volumes stored in the collector over the period between collections (Prechsl et al., 2014).

For end member mixing analysis (EMMA), samples were filtered with 0.45 µm polypropylene filters and collected in 25 to 30 ml HDPE bottles with screw cap. Samples were immediately acidified with nitric acid to pH<2 and stored frozen until analysis for trace elements Li, Na, Mg, Al, Si, K, Ca, Cr, Fe, Cu, Zn, Rb, Sr, Y, Ba, Ce, La and Nd with inductively coupled plasma mass spectrometry (ICP-MS) in the laboratory of Justus Liebig University Giessen, Germany ($n = 122$) or the University of

Hohenheim, Germany ($n = 231$). At the University of Hohenheim, samples were analysed for Al, Ca, K, Mg, Na and Si with inductively coupled plasma optical emission spectrometry (ICP-OES) instead of ICP-MS. Samples for EMMA were collected between 15 October 2015 and 21 October 2016. Weekly samples were taken for stream water, while precipitation and throughfall were sampled approximately every 4–6 weeks ($n = 9$–11). Due to difficult access to sampling sites, other potential water sources were sampled less frequently: wetland SHA-WL ($n = 4$) and spring NF-SP.b ($n = 3$). Springs NF-SP.a and TTP-

SP.a were a combination of samples taken at different locations rather than different points in time with $n = 2$ and $n = 5$, respectively. Ten shallow wells (SHA-WE.a and SHA-WE.b) in SHA were sampled twice. Initially all samples for this end member were combined, but SHA-WE.b showed a chemical composition that strongly differed from the remaining samples and was therefore treated as a separate end member. No separate end member sampling was carried out for OUT, except for one spring sample and regular precipitation samples. Since all end members from the sub-catchments were sampled within

OUT, these end members were used to identify potential streamflow sources for OUT. It was not possible to use samples collected from the wick samplers for EMMA, because the glass fibre wick could have contaminated the samples and the sample volume was generally too low (< 25 ml).

## 2.4    End member mixing analysis

The EMMA was carried out following the procedures described in Christophersen and Hooper (1992) and Hooper (2003). The

final set of solutes to be included in the EMMA was selected based on conservative behaviour of the solutes, which was assessed with bivariate scatter plots of all possible solute combinations, including stable water isotopes. A solute was considered conservative when it showed at least one significant ($p < 0.01$) linear relationship with another solute with $R^2 > 0.5$



(Hooper, 2003; James and Roulet, 2006). In our case these were Li, Na, Mg, K, Rb, Sr and Ba, i.e. elements which are commonly used in EMMA (Barthold et al., 2011).

The relative root mean square error (RRMSE) was calculated based on the measured and projected stream water concentrations for the selected solutes for up to four dimensions (i.e. principal components in EMMA). This was used to assess how many dimensions should be included in the analysis. Although higher-dimensional end member mixing models had lower RRMSE scores, the residual analysis (Hooper, 2003) and 'Rule of One' (last included dimension needs to explain at least $1/n^{th}$ of the variation, where $n$ is the number of solutes included in the analysis) both indicated that a 2-dimensional end member mixing model with three end members was sufficient for all catchments. Median end member concentrations were projected in the 2-dimensional mixing space of the stream water samples of the respective catchments and the three end members enclosing most of the stream water samples in this mixing space were selected for EMMA. Then, contributions of each end member to streamflow were calculated. Although it is common practice to project stream water samples that fall outside the triangle enclosed by the three selected end members back into the mixing space to constrain end member contributions to a range of 0 to 100 %, we decided to omit this step as it is indicative of uncertainty in the analysis caused by uncertainty in field and laboratory analyses, non-conservative solute behaviour, unidentified end members, and temporal variability of end members (Barthold et al., 2010).

### 2.5 Mean transit time analysis

#### 2.5.1 Model selection

Mean transit time (MTT) estimations of stream and soil water were obtained through lumped parameter models. In this approach, the transport of a tracer through a catchment is expressed mathematically by a convolution integral (Maloszewski and Zuber, 1982) in which the composition of the outflow (e.g. stream or soil water) $C_{out}$ at a time $t$ (time of exit) consists of a tracer $C_{in}$ that falls uniformly on the catchment in a previous time step $t'$ (time of entry), $C_{in}$ becomes lagged according to its transit time distribution $g(t-t')$. Having in mind that the time span $t-t'$ is in fact the tracer's transit time $\tau$, the convolution integral could be expressed as Eq. (1), in which $g(\tau)$ is the weighting function (i.e. the tracer's transit time distribution TTD) that describes the normalized distribution of the tracer added instantaneously over an entire area (McGuire and McDonnell, 2006).

$$C_{out}(t) = \int_0^\infty C_{in}(t - \tau) g(\tau) d\tau \qquad (1)$$

When using the convolution approach, any type of weighting function is referred as a lumped parameter model. In case preliminary insights of a system are to be obtained with scarce data, it is common practice to apply a set of models to analyse whether they yield similar results. Among the diverse model types, two-parameter models such as the gamma model (GM) or the exponential piston flow model (EPM) are commonly used for MTT estimations (Hrachowitz et al., 2010; McGuire and McDonnell, 2006) and were identified by Timbe et al. (2014) as most suited to infer MTT estimations of spring, stream and soil water in an Andean tropical montane forest catchment. We therefore chose to apply these models in our study (Table 2).



For EPM, the parameter $\eta$ is the ratio of the total volume to the volume of water with exponential distribution of transit times. If $\eta = 1$, the function corresponds to a fully exponential one-parameter model (EM), but there is no physical meaning for cases where $\eta < 1$. GM is a more general and flexible exponential-type of model. If $\alpha = 1$, the GM becomes an exponential model, but when $\alpha < 1$, a significant part of the flow is quickly transported to the river. Conversely, the signal of the concentration

peak is delayed for $\alpha > 1$.

The selection of acceptable model parameters was based on the statistical comparison of 50 000 random simulations (Monte Carlo approach), which assumes a uniform random distribution of the variables of each model. For each site and model, the performance was evaluated based on the best matches to a predefined objective function: the Nash-Sutcliffe efficiency (NSE). Quantification of errors and deviations from the observed data were calculated using the root mean square error (RMSE) and

the bias, respectively. MatLab R2017a was used for data handling and solving the convolution equation, while R was used for weighting the range of behavioural solutions (generalised likelihood uncertainty estimation, GLUE). When using GLUE, the range of behavioural solutions is discrete. In our case, the lower limit was set to 5 % below the best fitting efficiency. In order to refine the limits of behavioural solutions, the 90 % of the prediction limits were calculated for every variable through weighted quantiles between 0.05 and 0.95.

**2.5.2    Selection of isotope data for the MTT analyses**

Only $\delta^{18}O$ was used for MTT analysis, because the two measured conservative isotopes ($\delta^{18}O$ and $\delta^{2}H$) showed a strong linear relationship, meaning that similar estimations could be obtained by using just one isotope (Mosquera et al., 2016a). The isotopic signals of precipitation (weekly scheme, $n = 75$) were considered as input function of the lumped parameter models. The isotopic composition of throughfall samples, which were also collected (data not presented here) were not significantly

different from that of precipitation, hence the same MTT could be obtained using data from throughfall samples. All the available weekly isotope data for stream water ($n = 75$) were included in the analysis, because the seasonal isotopic signatures of stream water (i.e., TTP-RV, SHA-RV, NF-RV and OUT-RV) were considerably damped compared to the seasonal isotopic signatures of rainfall (Fig. 4). This means that, although some of the stream water samples could have been taken during interflow or high flow conditions, the isotopic signatures of those samples still showed a major component of 'old' or baseflow

water.

The number of soil water samples ($n = 4–47$) was smaller than for stream water ($n = 75$). This was because wick samplers – the devices used to collect soil water – only collect the portion of the water moving through the soil, i.e. they start to collect water for soil conditions near to saturation. Only three sites had enough data to perform model calibration and were therefore considered: NF-S15 ($n = 47$), OUT-S15 ($n = 47$) and OUT-S50 ($n = 46$).





## 3 Results

### 3.1 Solute concentrations

Most end members and stream water showed differences in median solute concentrations (Fig. 2). Especially samples from shallow well WE.b in the smallholder agriculture catchment (SHA) had higher concentrations for most solutes than other end members. Concentrations were lowest in precipitation (PC) in all catchments, while throughfall (TF) in some catchments showed higher concentrations and more variation. These patterns were reflected in the total solute concentrations of the different end members, although the difference between shallow well WE.b and the other end members was not as pronounced.

### 3.2 Isotopic composition

Isotopic values for precipitation plotted slightly above the global meteoric water line (GMWL), resulting in a local meteoric water line (LMWL) with a slope of 8.05±0.21 $\delta^{18}$O and an intercept of 15.31±0.61 $\delta^2$H ($p < 0.001$, $R^2 = 0.962$; Fig. 3). The slopes of the LMWL and GMWL were not significantly different ($p = 0.619$), but the intercepts were ($p < 0.001$). Samples far below the LMWL represented samples with a low sample volume ($< 100$ ml) affected by evaporative enrichment. There was no significant effect of elevation on $\delta^{18}$O values of the precipitation samples, but precipitation samples collected at higher altitude (SHA-PC) were generally more depleted than those collected at lower altitudes (NF-PC, TTP-PC and OUT-PC). The linear regression slope for stream water samples was 5.00±0.54 $\delta^{18}$O, which was significantly lower than the slope of the LMWL ($p < 0.001$). There was very little variation in isotopic values in streamflow throughout the study period, while values for precipitation showed pronounced minima in November 2015, May 2016 and November 2016 in all catchments (Fig. 4).

### 3.3 End member contributions

Based on the projection of all end members in the stream water mixing space for each catchment, it was possible to identify three end members that would enclose most of the stream water samples for NF and SHA (Fig. 5). However, this involved selection of two very specific sources with a low number of samples ($n = 2$), i.e. a combination of two springs NF-SP.a located close to each other, sampled on the same day for NF, and two samples taken from shallow well SHA-WE.b in the smallholder area. The sampled end members were not sufficient to capture the variability in stream water samples in TTP and OUT, with more than a third of the stream water samples falling outside the area enclosed by the three selected end members. Precipitation in all catchments plotted similarly in the mixing space of OUT. Also springs OUT-SP.b and NF-SP.b and the combination of nine shallow wells SHA-WE.a, as well as springs TTP-SP.a and NF-SP.a and wetland SHA-WL were similar, whereas there was considerable variation in chemical composition of throughfall (TF) samples, both within and between sub-catchments. Shallow well SHA-WE.b plotted far outside the mixing space of NF, TTP and OUT.

Predicted stream water solute concentrations, based on median solute concentrations of the selected end members, matched well with observed stream water solute concentrations ($R^2 > 0.85$ for most solutes). The EMMA resulted in a dominant contribution of precipitation (PC) in NF (45 %) and SHA (59 %), while spring water (TTP-SP.a) dominated in TTP (56 %)



(Fig. 6). The three selected end members for OUT generally had similar contributions (30–40 %). In NF and OUT the contribution of precipitation dropped towards the end of the dry season from more than 50 % to less than 10 % (March–April) and increased again to around 25 % during the long rains. In this period, the contribution of throughfall was higher in NF (62 %) and OUT (65 %). Conversely, in SHA a strong drop in contribution of precipitation (from 86 to 30 %) was observed at the start of the long rains in May 2016. Precipitation did not contribute to streamflow in TTP during the dry season, whereas the contribution of spring water TTP-SP.a was highly overestimated (up to 853 %). Contributions of end members during the second half of the study period in SHA differed from the first half, with an increase in contributions of wetland SHA-WL from 1 to 58 %. Generally, the contribution of the wetland was higher during periods of high flow in SHA (51 %) – similar to contributions of springs SP.a in NF and TTP. Conversely, shallow well SHA-WE.b in SHA showed highest contributions during the dry season (up to 54 %).

### 3.4 MTT estimates for stream and soil water

Based on the Nash-Sutcliffe efficiency (NSE), it was clear that the gamma model (GM) provided a better mean transit time (MTT) estimate for stream water than the exponential piston flow model (EPM; Table 3). The best performance was observed for OUT-RV (NSE = 0.33), while TTP-RV had a very low performance (NSE = 0.05) and was therefore discarded. The generally low fitting efficiencies were caused by the low amplitude of seasonal isotopic signatures of $\delta^{18}O$ in stream water samples from all four catchments (see standard deviation of observed values in Table 3; Fig. 3–4). There was a moderate positive relationship between the standard deviation of the observed values and corresponding NSE of modelled results ($R^2$ = 0.84). NF-RV and SHA-RV had a similar estimated MTT of approximately 4 years (Table 3). However, similar to TTP-RV, the poor fit to the objective functions (NSE = 0.15 and NSE = 0.22, respectively) could be related to the highly damped isotopic signature and should be interpreted with care. The shortest estimated MTT of 2.5 years was for OUT-RV.

For soil water, both models (GM and EPM) yielded similar results in terms of fitting efficiencies (NSE), MTT estimations and uncertainty ranges (Table 4). NF-S15 showed the shortest estimated transit time (3.2–3.3 weeks). The estimated transit time for OUT-S15 (4.5–7.5 weeks) was longer than for NF-S15, but shorter than for OUT-S50 (10.4–10.8 weeks).

## 4 Discussion

### 4.1 Hydrochemistry

While precipitation (PC) had low solute concentrations at all sites, throughfall (TF) concentrations were much more variable in space and time, although solute concentrations were generally not significantly different between sites. This has been observed elsewhere as well (e.g. Ali et al., 2010; Germer et al., 2007) and can be attributed to seasonal variations in plant growth and dry and wet atmospheric deposition of elements such as Na, K and Mg originating from sea salts or biomass burning. Shallow well SHA-WE.b had trace element concentrations that were much higher than those of the other nine sampled shallow wells SHA-WE.a. Wetland SHA-WL, located near shallow well SHA-WE.b, did not show these high concentrations,



which could indicate that the shallow well received water from a different groundwater source than the wetland and other shallow wells. Similarity in solute concentrations in springs NF-SP.b and OUT-SP.b and shallow wells SHA-WE.a indicate that these end members represent the same water source, despite their different geographical location. The same was observed for wetland SHA-WL and springs NF-SP.a and TTP-SP.a.

The higher intercept of the local meteoric water line (LMWL) than of the global meteoric water line (GMWL) indicates deuterium-excess (*d*-excess) as consequence of more arid vapour sources (McGuire and McDonnell, 2007) or re-evaporated rainfall (Goldsmith et al., 2012). Similar *d*-excess values have been observed in many tropical montane environments (e.g. Goldsmith et al., 2012; Mosquera et al., 2016a; Muñoz-Villers et al., 2016; Otte et al., 2017; Windhorst et al., 2013). The value for the slope of the linear relation between stream water isotopic values (5.00±0.54) was similar to that found by Craig (1961)

for East African rivers and lakes and suggests evaporative enrichment of stream water. The observed altitude effect ($-0.099$ ‰ $\delta^{18}O$ per 100 m) is smaller than the $-0.22$ ‰ $\delta^{18}O$ per 100 m found in an Andean tropical montane forest (Windhorst et al., 2013), $-0.31$ ‰ $\delta^{18}O$ per 100 m in an Ecuadorian Páramo ecosystem (Mosquera et al., 2016a), but similar to values of $-0.10$ and $-0.11$ ‰ $\delta^{18}O$ per 100 m observed on Mt. Kilimanjaro in Tanzania (Mckenzie et al., 2010; Otte et al., 2017). The occurrence of the lowest precipitation $\delta^{18}O$ values during the rainy seasons also agrees with seasonal observations by Otte et

al. (2017) on Mt. Kilimanjaro and is most likely related to the different isotopic composition of precipitation from storms caused by the movement of the intertropical convergence zone (ITCZ) over the study area during the rainy seasons (Otte et al., 2017). Furthermore, most storm trajectories originate from south-easterly direction during the long and short rainy season, while coming from an easterly direction during the dry season, suggesting different origin and thus isotopic composition of precipitation (Soderberg et al., 2013).

**4.2   Dominant water sources**

The end member mixing analysis (EMMA) showed that precipitation (PC) was always one of the three selected end members in all catchments, as depicted in our conceptual model of the rainfall–runoff generation processes in the three sub-catchments with different land use (Fig. 7). Although the use of a single throughfall sampler might not be sufficient to capture the spatial variation in throughfall chemistry (Zimmermann et al., 2007), throughfall (TF) was selected as an additional end member for

all catchments, except in the smallholder agriculture sub-catchment (SHA). The high contribution of precipitation (21–59 %) in all catchments and throughfall (31–40 %) in the natural forest (NF) sub-catchment and the main catchment (OUT) suggest high contributions of channel precipitation, surface runoff or rapid sub-surface flow. However, given the size of streams, it is unlikely that channel precipitation alters the stream's composition to such an extent. Although surface runoff can occur in tropical forests (e.g. Chaves et al., 2008; Johnson et al., 2006; de Moraes et al., 2006) and was observed on paths in NF, a

major contribution of surface runoff is unlikely due to high infiltration rates and hydraulic conductivity of forest soils (Owuor et al., 2018). We therefore conclude that the observed signatures were caused by shallow sub-surface flow during rainfall events, which agrees with findings in NF by Jacobs et al. (in review) and is commonly observed in tropical montane forested catchments (e.g. Boy et al., 2008; Muñoz-Villers and McDonnell, 2012; Saunders et al., 2006). The extent to which the





chemical composition of water changes through contact with the soil depends on the contact time (McGuire and McDonnell, 2006; Mulholland et al., 1990). Therefore, if event water, i.e. precipitation or throughfall, is only in contact with the soil for a short time (e.g. several hours), the chemical composition of the water that enters the stream might be comparable to the composition of precipitation or throughfall. Furthermore, if the riparian zone is near saturation, which occurs in the relatively

flat valley bottoms in NF, only a small fraction of the precipitation can infiltrate and storage capacity is limited, resulting in shallow flow from the riparian zone during rainfall events (von Freyberg et al., 2014; Mosquera et al., 2015). Similar to our study, Chaves et al. (2008) found that the precipitation/throughfall end member contributed most to streamflow in a forested Amazonian catchment.

The relatively low contribution of precipitation to streamflow in the tea and tree plantation sub-catchment (TTP) compared to

the other sub-catchments suggests a minor input of surface runoff to streamflow during both wet and dry conditions (Fig. 7). This seemingly contradicts previous findings in the same sub-catchment, where rainfall events led to significant dilution of nitrate concentrations in stream water due to surface runoff (Jacobs et al., in review). However, surface runoff could have a different chemical signature than precipitation (Chaves et al., 2008). Most of the surface runoff in TTP seems to be generated on footpaths and roads. Emissions from traffic and wear of tyres could also change the surface runoff composition (Gan et al.,

2008). However, the chemical composition of stream water samples did not correspond to trace elements related to traffic (Mn, Pb, Cu, Zn and Cr; Gunawardena et al., 2015), but rather indicated mineral origin (high concentrations of Si, Li, K, Na and Rb; data not presented here). Specific sampling of surface runoff and subsequent inclusion as separate end member could improve the end member mixing model performance. Similar to Muñoz-Villers and McDonnell (2012) and Chaves et al. (2008), the contribution of precipitation and throughfall decreased in all sub-catchments during high flows (Fig. 7, right

hillslopes in each graph). This suggests increased inputs from groundwater through wetlands (SHA-WL) or springs (TTP-SP.a and NF-SP.a) during the rainy season. These findings support our hypothesis that there are temporal changes in the contribution of the different end members in this African tropical montane ecosystem, similar to South American tropical montane catchment (Chaves et al., 2008; Correa et al., 2017). Groundwater end member SHA-WE.b in SHA showed contrasting behaviour, with highest contributions during low flow periods, suggesting that this is a different groundwater source and an

important component of baseflow in SHA.

The triangle bounded by the three selected end members in the stream water mixing space of NF (precipitation, throughfall and springs SP.a; Fig. 5) encompassed most of the stream water samples, with only 9 % of the samples falling outside the triangle. However, in SHA, TTP and OUT 42, 49 and 33 % of the samples fell outside the triangle of the three selected end members, respectively. Although this could be attributed to the variability in end member composition, uncertainty in

laboratory analysis or non-conservative solute behaviour (Barthold et al., 2010), it is very likely that one or more end members are missing, which could be better suited to explain the observed chemical composition of stream water at the catchment outlet. Alternatively, inclusion of additional end members to increase dimensionality of the end member model may be required to satisfactorily represent the behaviour and stream water sources in these catchments, as observed for an Andean Páramo ecosystem (Correa et al., 2017). The selection of tracers and number of end members is highly subjective and can therefore





significantly affect the outcomes of the EMMA (Barthold et al., 2011). Furthermore, although the chemical signature of end members should be invariable in space and time according to the EMMA assumptions, a more consistent sampling approach whereby all end members are sampled on a regular basis could also improve the performance of the models, because the full range of chemical variation in time would be captured (Neill et al., 2011). In our case this was not possible, because most

sampling sites were difficult to access.

Another shortcoming of our sampling approach is that springs, shallow wells and wetlands might not accurately represent groundwater, although this could be an important end member, as observed in many studies (e.g. Barthold et al., 2011; Chaves et al., 2008; Crespo et al., 2012; Katsuyama et al., 2009). Access to groundwater in the study area is complicated by the absence of wells or boreholes in NF and TTP, and the existing wells in SHA are often not properly sealed, which means that

groundwater can mix with water from shallower soil layers and precipitation, obscuring the groundwater signal. Jacobs et al. (in review) suggested that discharge contributing zones change with the seasons, which could be tested by inclusion of soil water as end member. This was not possible with the current experimental set-up, because the glass fibre wick in the wick samplers could contaminate the trace element samples. Especially soil water from different topographical locations within the catchment (e.g. riparian zone and hillslope) or different soil types could yield further insight in the dynamics of discharge

contributing zones and important flow paths during different seasons.

### 4.3     Mean transit times

The low variation in isotopic signatures ($-3.6$ to $-0.3$ ‰ for $\delta^{18}O$) observed for stream water compared to precipitation ($-9.9$ to $4.4$ ‰) at all sites suggests long travel times. Equally damped signals ($-8.0$ to $-6.2$ ‰ versus $-15.2$ to $-0.4$ ‰ for $\delta^{18}O$ in stream water and precipitation, respectively) were observed in a Mexican tropical montane forest catchment (Muñoz-Villers

and McDonnell, 2012). The long transit time could be explained by the deep and well-drained soils in our study area (Cooper, 1979; Edwards and Blackie, 1981), which promote slow flow paths through deeper soil layers and longer transit times (Asano and Uchida, 2012). The most damped isotopic signature was observed at TTP (SD = 0.26 ‰ for $\delta^{18}O$; Table 3), which suggests that stream water in this sub-catchments is older than at all other sites. Most likely, the MTT is longer than 4 years and is therefore beyond the reliability of the present used method with $\delta^{18}O$ or $\delta^{2}H$ tracers, which also explains the very low Nash-

Sutcliffe efficiency (NSE). Better predictions could be obtained by using more appropriate tracers for estimating transit times of several years to decades, such at tritium ($^{3}H$) (Cartwright et al., 2017). A longer sampling period of at least 4 years would also improve the reliability of the mean transit time estimates (McGuire and McDonnell, 2006). Although the gamma model (GM) used in this study was found to be most suitable for the estimation of stream water MTT in other tropical montane catchments (Muñoz-Villers and McDonnell, 2012; Timbe et al., 2014), it is also possible that the applied method for MTT

estimation is less suitable for tropical catchments with highly damped isotope signals and low seasonal variation, as indicated by the low NSE for all stream water sites.

Because of the similar estimated MTTs for NF and SHA and the most likely longer MTT for TTP, we rejected our hypothesis that agricultural catchments have a shorter MTT than forested catchments due to increased importance of faster flow paths





such as surface runoff. Evidence from other studies suggests that the role of vegetation cover in water storage and MTT could be suppressed by geomorphology (Timbe et al., 2017) or soil hydraulic properties (Geris et al., 2015; Mueller et al., 2013; Muñoz-Villers et al., 2016). The latter, however, can also be influenced by land use. The MTT of ~4 years in the three sub-catchments suggests that most of the stream water originates from 'old' water or groundwater, which corresponds with the

importance of groundwater-related end members springs TTP-SP.a and NF-SP.a and wetland SHA-WL in the sub-catchments. The runoff ratios in all catchments (0.323–0.387) confirm that a small part of the precipitation leaves the catchment as discharge. Similar runoff ratios (0.30) and MTT (~3 years) were obtained in a Mexican montane forest catchment with deep volcanic soils, but higher annual precipitation (Muñoz-Villers and McDonnell, 2012). However, Andean tropical montane catchments had higher runoff ratios (0.76–0.81) and correspondingly shorter MTTs (<1 year) (Crespo et al., 2012), which

could be caused by steeper slopes and shallower soils compared to our study area. The importance of groundwater does, however, contradict the generally high contribution of precipitation and throughfall to streamflow in most catchments. The use of bulked precipitation and weekly stream water samples as input could cause a bias towards older groundwater, because the direct effect of storm events on stream water isotope composition are removed from the analysis (McGuire and McDonnell, 2006). Although samples obtained during high flow in the rainy season were not removed from our analysis, the use of bulked

samples could have underestimated the importance of faster flow paths during rainfall events and therefore partly explain the discrepancy between the long transit times and high contribution of precipitation and throughfall to streamflow in most catchments.

The shorter estimated MTT for OUT compared to the sub-catchments is counterintuitive, since it is the largest catchment. One could also expect that, since OUT is a mixture of the three land use types dominating the sub-catchments, the MTT should be

similar to or an average of the estimated MTTs of the sub-catchments. MTT is, however, not always correlated to catchment area (McGuire et al., 2005; Rodgers et al., 2005), but seems more related to other hydrological and topographical metrics such as drainage density and slope (Capell et al., 2012). Also geology and presence of hydrologically responsive soils seem to be important determinants for MTT (Capell et al., 2012; Tetzlaff et al., 2007). The occurrence of other soil types (mollic Andosols) and underlying geology (pyroclastic unconsolidated rock) in the upper part of OUT (ISRIC, 2007) compared to the humic

Nitisols and igneous rock dominating the three sub-catchments could lead to differences in soil hydraulic properties and sub-surface water storage and eventually MTT, but not enough data are available for the study area to test this.

The longer MTT for soil water for OUT-S15, located in a pasture, than for NF-S15 contradicts findings by Timbe et al. (2014), who compared pasture and forest soil water MTT and found longer MTTs for forested sites. In our case, the difference could be caused by differences in hydraulic conductivity. Pasture soils in our study area had a generally lower hydraulic conductivity

(2–53 cm h$^{-1}$) than natural forest soils (10–207 cm h$^{-1}$) due to soil compaction by animal trampling (Owuor et al., 2018). Differences in soil hydraulic properties between land use types are, however, mainly restricted to the topsoil, while deeper soil layers are usually less affected by land management (Zimmermann et al., 2006). The estimated MTTs fell within the range observed for soil water from 30 to 60 cm depth (20–62 days) in a tropical montane catchment in Mexico (Muñoz-Villers and McDonnell, 2012). For soil water MTT estimation, the second parameter ($\alpha$) for GM was around 1.5 for the best-modelled

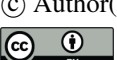



efficiencies for NF-S15 and OUT-S50. However, according to the range of behavioural solutions, all soil sites could be well represented by gamma functions with $\alpha$ values of 1 (Table 4), which means a simple exponential distribution function (EM). Similarly, the exponential piston flow model (EPM) can yield similar results with $\eta = 1$ for all analysed cases, meaning that EPM could also be simplified as EM, i.e. without any portion of piston flow participating in the transport. Therefore, results

from both models point out that the same predictions could be obtained with a simpler, single parameter exponential model, as was used for estimation of MTT of soil water at 30 cm by Muñoz-Villers and McDonnell (2012). In order to avoid over-parametrization, models with less parameters are preferred when they provide comparable results.

## 5    Conclusion

In this study we aimed to identify the dominant water sources and flow paths in three sub-catchments with contrasting land

use (i.e. natural forest, smallholder agriculture and commercial tea and tree plantations) using mean transit time (MTT) analysis and end member mixing analysis (EMMA) to assess the effect of land use on catchment hydrology. The analyses revealed a similar MTT of approximately 4 years in all catchments, which is longer than observed in other tropical montane headwater catchments. In the three sub-catchments, springs and wetlands fed by groundwater were selected as important end member, with increased contribution to streamflow during high flows. A second, different groundwater source was identified in the

smallholder agriculture catchment, which was an important end member during baseflow. These results emphasize the importance of sufficient groundwater recharge and sustainable management of groundwater resources to maintain streamflow throughout the year.

Despite the observed similarities, the three sub-catchments showed clear differences in the contribution of precipitation and throughfall to stream water, with highest contributions in the natural forest and smallholder agriculture and lowest contribution

in the tea and tree plantations. However, we expect that the contribution of precipitation and throughfall in the natural forest sub-catchment occurs as shallow sub-surface flow, while surface runoff could still play a significant role in the smallholder agriculture sub-catchment. Further evidence to support this statement is necessary, because surface runoff generally has negative impact on soil fertility, erosion and sedimentation. Due the similar soils and geology in the three sub-catchments, the differences in end member selection and behaviour can mainly be attributed to land use. However, over- and under-prediction

of end member contributions, especially during the dry season and at the peak of the rainy season, indicate that important end members were missing in the mixing models. Identification of additional end members and regular sampling of all end members to capture the variation in chemical composition of the end members throughout the year, might therefore improve the end member mixing models and thus our knowledge on dominant water sources and flow paths in the three land use types under different hydrological regimes. Because changes in flow paths will affect the transport and fate of nutrients and

pollutants, which could have an adverse effect on montane ecosystems and downstream areas, the results of this study can be used to assess the potential impact of future land use changes on surface water supply and quality.



**Data availability**

Hydroclimatic data (discharge and precipitation) and the full isotope and trace element dataset for all study sites is available from the online database http://fb09-pasig.umwelt.uni-giessen.de:8050/wiki/publications hosted by Justus Liebig University, Giessen, Germany.

**Author contributions**

The study was designed by SJ, BW and LB. SJ and BW installed all instruments. SJ was in charge of field campaigns, instrument maintenance and sample collection, and performed end member mixing analysis. BW managed the laboratory analysis. ET performed the analysis for mean transit time estimation. SJ, MR, KBB and LB prepared the manuscript.

**Competing interests**

The authors declare that they have no conflict of interest.

**Acknowledgements**

We would like to thank the Kenya Forest Service (KFS) for supporting us to conduct this study in the South-West Mau. This work was partially funded by the CGIAR program on Forest, Trees and Agroforestry led by the Centre for International Forestry Research (CIFOR). We thank the Deutsche Forschungsgemeinschaft DFG (BR2238/23-1) and the Deutsche
Gesellschaft für Internationale Zusammenarbeit GIZ (Grants 81195001 "Low cost methods for monitoring water quality to inform upscaling of sustainable water management in forested landscapes in Kenya") for generously providing additional support.

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



**Figure 1.** Map of the study area in the South-West Mau, Kenya, showing the three sub-catchments with different land use types within the main catchment, location of rain gauges, and sampling sites for stream water and selected end members. Sampling sites with overlapping symbols are indicated with labels instead of symbols. Numbers in brackets in the legend indicate the number of sampling sites per end member.





**Figure 2.** Box plots with concentrations of (a) Li, (b) Na, (c) Rb, (d) Mg, (e) Sr, (f) K and (g) Ba, and (h) total concentration of the selected solutes in stream water and sampled end members in the three sub-catchments with different land use (NF = natural forest, SHA = smallholder agriculture, TTP = tea and tree plantations) and the main catchment (OUT) between 15 October 2015 and 21 October 2016 in the South-West Mau, Kenya. The thick line represents the median, the box shows the interquartile range and the whiskers the minimum and maximum values within 1.5 times the interquartile range. Outliers are indicated with open circles. Numbers in plot (h) indicate the number of samples per end member.





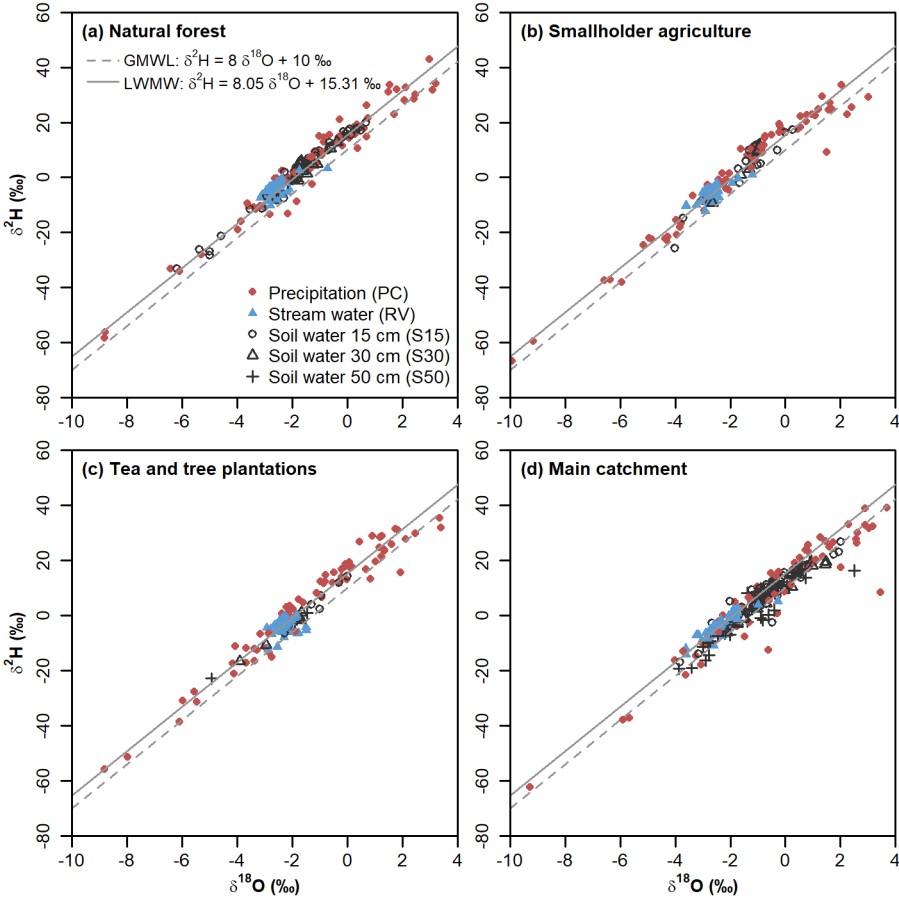

**Figure 3.** Relationship between $\delta^{18}O$ and $\delta^{2}H$ values in precipitation (PC), stream water (RV) and soil water at 15, 30 and 50 cm depth (S15, S30 and S50, respectively) for the (a) natural forest (NF), (b) smallholder agriculture (SHA), and (c) tea and tree plantations (TTP) sub-catchments, and (d) the main catchment (OUT) between 15 October 2015 and 17 March 2017 in the South-West Mau, Kenya. The global meteoric water line (GMWL) and local meteoric water line (LMWL) are indicated as dashed and solid lines, respectively.





**Figure 4.** Time series of $\delta^{18}$O values in precipitation (PC), stream water (RV) and soil water at 15, 30 and 50 cm depth (S15, S30 and S50, respectively), specific discharge and weekly precipitation in the (a) natural forest (NF), (b) smallholder agriculture (SHA), and (c) tea and tree plantations (TTP) sub-catchments, and (d) the main catchment (OUT) between 15 October 2015 and 17 March 2017 in the South-West Mau, Kenya.





**Figure 5.** Projection of end members in the 2-dimensional (U1 and U2) mixing space of stream water samples of the (a) natural forest (NF), (b) smallholder agriculture (SHA), and (c) tea and tree plantation (TTP) sub-catchments and (d) the main catchment (OUT) between 15 October 2015 and 21 October 2016 in the South-West Mau, Kenya. The size of the symbol for stream water represents the relative discharge at the time of sampling (larger symbol means higher discharge).





**Figure 6.** Specific discharge (shaded) and contribution of selected end members to streamflow for the (a–b) natural forest (NF), (c–d) smallholder agriculture (SHA) and (e–f) tea and tea plantation (TTP) sub-catchments and (g–h) the main catchment (OUT) between 15 October 2015 and 21 October 2016 in the South-West Mau, Kenya. The grey dashed lines indicate the realistic range of end member contributions and arrows show sampling dates for end members. The thick line in the box plots represents the median, the box shows the interquartile range and the whiskers the minimum and maximum values within 1.5 times the interquartile range. Outliers are indicated with open circles.





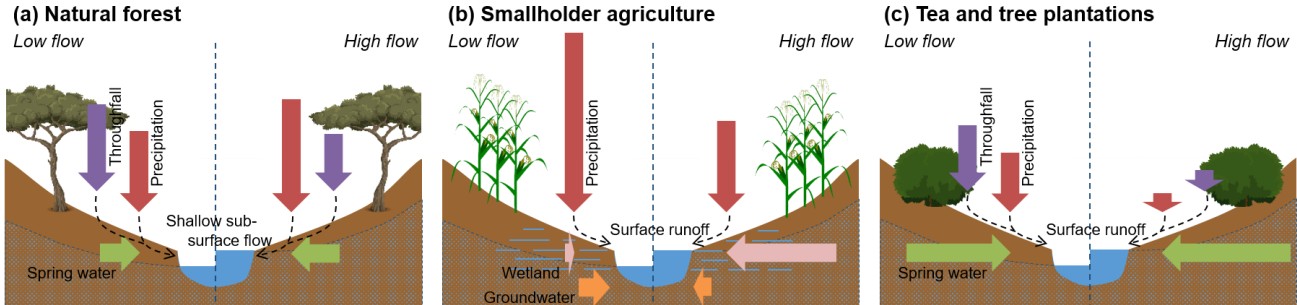

**Figure 7.** Conceptual model of dominant water sources and flow paths in different land use types during low (≤ mean discharge) and high flows (> mean discharge) in a tropical montane area: (a) natural forest (NF), (b) smallholder agriculture (SHA) and (c) commercial tea and tree plantations (TTP), based on results of end member mixing and mean transit time analysis in the South-West Mau, Kenya. Arrow length represents the median contribution (%) of each end member. Black dashed arrows show the most likely pathway for precipitation and throughfall to reach the stream.



**Table 1.** Physical and hydroclimatic characteristics of the study catchments in the South-West Mau, Kenya. Precipitation, specific discharge and runoff ratio are presented for the study period of 15 October 2015 to 14 October 2016.

| Catchment | Area | Elevation | Slope[a] | Precipitation | Specific discharge | RR[b] |
|---|---|---|---|---|---|---|
| | $km^2$ | $m$ | $\%$ | $mm\ yr^{-1}$ | $mm\ yr^{-1}$ | - |
| Natural forest (NF) | 35.9 | 1 954–2 385 | 15.5±8.0 | 2 299 | 744 | 0.323 |
| Smallholder agriculture (SHA) | 27.2 | 2 380–2 691 | 11.5±6.5 | 1 738 | 607 | 0.349 |
| Tea and tree plantations (TTP) | 33.3 | 1 786–2 141 | 12.2±7.3 | 2 045 | 791 | 0.387 |
| Main catchment (OUT) | 1021.3 | 1 715–2 932 | 12.8±7.7 | 2 019 | 701 | 0.347 |

[a] Mean±SD; [b] Runoff ratio, i.e. ratio of specific discharge to precipitation



**Table 2.** The lumped parameter models used for the estimation of mean transit times in the South-West Mau, Kenya.

| Model | Transit time distribution $g(\tau)$ | Parameter range for Monte Carlo simulations[a] |
|---|---|---|
| Gamma model (GM) | $\dfrac{\tau^{\alpha-1}}{\beta^{\alpha}\Gamma(\alpha)}\exp\left(-\dfrac{\tau}{\beta}\right)$ | $\alpha$ [0.0001–10] <br> $\tau$ [1–400] <br> $\beta = \alpha/\tau$ |
| Exponential piston flow model (EPM) | $\dfrac{\eta}{\tau}\exp\left(-\dfrac{\eta t}{\tau}+\eta-1\right)$ for $t \geq \tau(1-\eta^{-1})$ <br><br> $0$ for $t < \tau(1-\eta^{-1})$ | $\tau$ [1–400] <br> $\eta$ [0.1–4] |

[a] $\tau$ = tracer's mean transit time; $\alpha$ and $\beta$ = shape parameters; $\eta$ ratio of the total volume to the volume of water with exponential distribution of transit times. Units for parameters and their respective ranges are a-dimensional except for $\tau$, which has units of time.





**Table 3.** Main statistical parameters of observed and modelled $\delta^{18}$O for stream water in the three sub-catchments and the main catchments for the gamma model (GM) and exponential piston flow model (EPM). Uncertainty bounds of the modelled parameters ($\tau$ and $\alpha$ or $\eta$), in parentheses, were calculated through generalized likelihood uncertainty estimation (GLUE).

| Site[a] | Area | Elevation | Observed $\delta^{18}$O | | Modelled $\delta^{18}$O | | | | | | |
|---|---|---|---|---|---|---|---|---|---|---|---|
| | | | Mean | SD[b] | Mean | SD[b] | NSE[c] | RMSE[d] | Bias | MTT[e] | $\alpha/\eta$[f] |
| | $km^2$ | $m$ | ‰ | ‰ | ‰ | ‰ | - | ‰ | ‰ | $years$ | - |
| *Gamma model (GM)* | | | | | | | | | | | |
| NF-RV | 35.9 | 1 969 | −2.58 | 0.32 | −2.56 | 0.13 | 0.15 | 0.30 | 0.021 | 4.0 (3.3–4.6) | 0.65 (0.63–0.71) |
| SHA-RV | 27.2 | 2 386 | −2.72 | 0.31 | −2.69 | 0.16 | 0.22 | 0.27 | 0.029 | 3.8 (3.1–4.5) | 0.61 (0.57–0.66) |
| TTP-RV | 33.3 | 1 788 | −2.29 | 0.26 | −2.29 | 0.06 | 0.05 | 0.25 | 0.000 | 3.3 (2.8–4.3) | 1.09 (0.99–1.17) |
| OUT-RV | 1021.3 | 1 717 | −2.42 | 0.47 | −2.36 | 0.26 | 0.33 | 0.38 | 0.061 | 2.5 (1.8–3.4) | 0.48 (0.43–0.54) |
| *Exponential piston flow model (EPM)* | | | | | | | | | | | |
| NF-RV | 35.9 | 1 969 | −2.58 | 0.32 | −2.58 | 0.10 | 0.09 | 0.31 | 0.000 | 2.4 (2.1–2.9) | 1.000 (0.994–1.003) |
| SHA-RV | 27.2 | 2 386 | −2.72 | 0.31 | −2.72 | 0.11 | 0.12 | 0.29 | 0.000 | 2.2 (1.9–2.6) | 1.001 (0.994–1.004) |
| TTP-RV | 33.3 | 1 788 | −2.29 | 0.26 | −2.29 | 0.07 | 0.07 | 0.25 | 0.000 | 3.5 (3.1–4.1) | 1.011 (1.009–1.018) |
| OUT-RV | 1 021.3 | 1 717 | −2.42 | 0.47 | −2.42 | 0.20 | 0.14 | 0.43 | 0.001 | 1.2 (1.0–1.4) | 1.001 (0.998–1.007) |

[a] NF = natural forest, SHA = smallholder agriculture, TTP = tea and tree plantations, OUT = main catchment, RV = stream

5   water; [b] standard deviation; [c] Nash-Sutcliffe efficiency of objective function; [d] root mean square error; [e] estimated mean transit

time (in years); [f] model parameters for GM ($\alpha$) and EPM ($\eta$).



**Table 4.** Main statistical parameters of observed and modelled $\delta^{18}O$ for soil water at 15 cm depth in the natural forest sub-catchment and at 15 and 50 cm depth in the main catchment for the gamma model (GM) and exponential piston flow model (EPM). Uncertainty bounds of the modelled parameters ($\tau$ and $\alpha$ or $\eta$), in parentheses, were calculated through generalized likelihood uncertainty estimation (GLUE).

| Site[a] | $n$[b] | Elevation | Observed $\delta^{18}O$ | | Modelled $\delta^{18}O$ | | | | | | |
|---|---|---|---|---|---|---|---|---|---|---|---|
| | | | Mean | SD[c] | Mean | SD[c] | NSE[d] | RMSE[e] | Bias | MTT[f] | $\alpha/\eta$[g] |
| | - | m | ‰ | ‰ | ‰ | ‰ | - | ‰ | ‰ | weeks | - |
| *Gamma model (GM)* | | | | | | | | | | | |
| NF-S15 | 47 | 1 971 | −1.62 | 1.64 | −1.74 | 1.48 | 0.79 | 0.75 | −0.12 | 3.2 (2.8–4.1) | 1.5 (0.9–2.2) |
| OUT-S15 | 47 | 1 721 | −0.68 | 1.20 | −0.71 | 0.99 | 0.50 | 0.84 | −0.03 | 7.9 (6.1–11.3) | 0.9 (0.6–1.2) |
| OUT-S50 | 46 | 1 721 | −0.84 | 1.35 | −0.92 | 0.93 | 0.47 | 0.97 | −0.08 | 10.4 (8.8–12.6) | 1.4 (1.1–2.0) |
| *Exponential piston flow model (EPM)* | | | | | | | | | | | |
| NF-S15 | 47 | 1 971 | −1.62 | 1.64 | −1.67 | 1.38 | 0.78 | 0.77 | −0.05 | 3.3 (2.6–4.4) | 1.0 (0.9–1.1) |
| OUT-S15 | 47 | 1 721 | −0.68 | 1.20 | −0.58 | 0.94 | 0.52 | 0.82 | 0.11 | 4.5 (3.2–6.7) | 0.8 (0.7–1.1) |
| OUT-S50 | 46 | 1 721 | −0.84 | 1.35 | −0.90 | 0.85 | 0.46 | 0.99 | −0.06 | 10.8 (8.0–13.9) | 1.0 (0.9–1.3) |

[a] NF = natural forest, OUT = main catchment, S15 = soil water 15 cm depth, S50 = soil water 50 cm depth; [f] number of
5    samples; [c] standard deviation; [d] Nash-Sutcliffe efficiency of objective function; [e] root mean square error; [f] predicted mean transit time (in weeks); [g] model parameters for GM ($\alpha$) and EPM ($\eta$).