# Peer review of "Assessment of hydrological pathways in East African montane catchments under different land use"

_Hydrology and Earth System Sciences, 2018_

## Referee Comment (RC1) · Anonymous Referee #1 · 6 Mar 2018

The commented manuscript presents an exercise to analyse dominant sources and transit times for stream and soil mobile waters in a tropical montane catchment in East Africa, subject to diverse land uses.

General comments:

The subject of the manuscript may be of interest for HESS readers, it represents a relevant work volume and is well presented, but there are several formal and methodological issues that deserve a major revision of the manuscript before being acceptable for publication. The first issue is in the title of the paper itself. It is very assertive while the results of the work, taking into account the associated uncertainties, are much less

convincing. So, I suggest to change the title or just to put it into an interrogative form.

The second but main issue is respect to how Mean Transit Times (MTTs) have been obtained for stream waters. The several aspects of this issue are the following ones:

1) The MTT methodological explanation is adequate (if some citation of GLUE development papers is included) but it fails to describe how a (400?) year-long 18O input function has been obtained to feed the lumped models when the rainfall sampling period was just 75 weeks long.

2) It has been shown that MTT determinations using seasonal variations of tracer signals (such as the 18O one) cannot provide acceptable results longer than a few months in stream (mixed) waters due to the strong non-linearity of the driving function (Kirchner, 2016).

3) For such damped tracer signals in the stream waters and low model efficiencies, much larger MTT uncertainties should be obtained, showing results coherent with point 2. My opinion is that the small uncertainties obtained are an artefact due to the way the behavioural models have been selected in the GLUE exercise. Accepting only parameter sets with efficiency just 5% lower than the optimal one might be appropriate for high efficiency values, but not in the case of such low efficiency values because the range of behavioural parameters becomes too narrow. Some GLUE published works dealing with large uncertainties sensibly used all parameter sets with positive efficiencies. Alternatively, all the parameter sets with such low efficiencies might be rejected as a way to resolve that the method is inappropriate.

In the case of stream waters, I suggest to remove the proposed MTT determinations, unless the above points are adequately answered. The authors may reasonably continue using the clear damping of the tracer signal in the stream waters as an indicator of several-year old waters, and even the differences in the temporal variability of the tracer signals might be used to indirectly rank the waters MTTs. In the case of soil mobile waters, I suggest the application of some analysis of the significance of MTT differences

found, using the MTTs likelihood distributions provided by the GLUE exercise.

The third but also relevant issue refers to the End Member Mixing Analysis (EMMA) for the Small Holder Agriculture (SHA) stream waters. The use of the well SHA-WE.b as end member representative of groundwater chemistry is not reasonable. One well in the headwaters with solute concentrations very different from those in other nine wells may represent either a different water source or some pollution effect, but it is not sensible to hypothesize that it can be a relevant source for stream water when its chemistry is very local as it is not transmitted to the other well waters. The analysis done can be shown as a test, but it cannot be taken as representative because groundwater contribution becomes underestimated and the other components overestimated. If well understood, the use this end member with very low contributions as representative of groundwater is depicted in Figure 7 (b), although this is inconsistent with some text in the conclusions: "A second, different groundwater source was identified in the small-holder agriculture catchment, which was an important end member during baseflow"

Another more formal issue is the use of the 'soil water' expression to identify the samples of mobile waters sampled at different soil depths. In the current water isotope literature, 'soil water' refers to the total (bulk) water contained in the soil, including mobile and immobile waters. In the methods section it is clearly justified that just mobile water was sampled, but in the abstract, figures and conclusions, some adjective such as 'mobile' or 'free' should be added to 'soil water' in order to avoid any misunderstanding.

Detailed comments:

As most of the paper should be rewritten only major remarks not included before are made

- Page 3, line 21: some hypothesis on how rain water reaches the stream should be added

- Page 7, line 8: Nash & Sutcliffe (1970)

- Page 8, line 11: GLUE was first described in Beven & Binley (1992)

- Page 3, line 21: "Ten shallow wells (nine named SHA-WE.a and one SHA-WE.b)..."

- Page 8, lines 10 and 15: the units for the slopes are not correct.

- Page8, line 15; page 10 line 9: this slope value seems too small looking to the graphs.

- Page8, line 31: the contribution of precipitation to SHA stream waters is overestimated due to the role of SHA-WE.b commented above

References

Beven, K., & Binley, A. (1992). The future of distributed models: model calibration and uncertainty prediction. Hydrological processes, 6(3), 279-298.

Kirchner, J. W. (2016). Aggregation in environmental systems–Part 1: Seasonal tracer cycles quantify young water fractions, but not mean transit times, in spatially heterogeneous catchments. Hydrology and Earth System Sciences, 20(1), 279.

Nash, J. E., & Sutcliffe, J. V. (1970). River flow forecasting through conceptual models part I—A discussion of principles. Journal of hydrology, 10(3), 282-290.

---

## Referee Comment (RC2) · Anonymous Referee #2 · 13 Mar 2018

This study presents an isotope driven investigation of water cycling in a catchment in Kenya. Travel times and source apportionment for various land covers are considered. The study targets a gap in understanding centering around tropical montane landscapes. The study is well written and an appropriate topic for HESS. I have two general concerns about how the data are being presented and general manuscript structuring.

(1) Data presentation and analysis

There is some concern with taking these isotopic data into the convolution modeling. Specifically, the length of observation record is not overly long which limits the ability to

map out some realistic travel times here. The uncertainty gets very high in this regard. So, at best one could argue that the MTT estimations are just first-order assessments for comparing between the catchments. Further, taking on a classic time invariant MTT estimation is a bit troublesome with regard to the potential for comparisons. Namely, the travel times for each catchment likely shift with wetness (storage) condition and this dynamic shifting (mixing) likely represents itself differently throughout the one and a half year been considered here. As such, it is difficult to separate the impact of the land use on MTT from the variability of the flow on the MTT – more likely these aspects compound (and confound) the issue (see van der Velde et al., 2015, Consequences of mixing assumptions for time-variable travel time distributions, Hydrological Processes). Some consideration of these aspect must be taken up within the analysis – or at a minimum in the discussion with regards to impacts on the results and interpretations. As these estimates are currently presented they tend to over-sell the ability of such analysis and what we can truly learn from them. These MTT estimation techniques are far from perfect and difficult to connect with mechanisms. It would be unfortunately for the uncertainty inherent in them to conflate with our understanding of these sites.

Actually upon deeper reading, I am not sure the MTT analysis is truly justified or even needed in this study. The high uncertainty and extrapolation needed to make the convolution effort make sense and to interpret the results are not justified. This study would be more powerful to be a data presentation with an EMMA analysis constrained by the uncertainty inherent in these data which were hard to collect. The current MTT analysis is just too thinly supported by the minimal data and has no real consideration of variability versus uncertainty to allow for a rigorous interpretation. I would strongly recommend removing these parts of the study and focusing in on the other aspects to make for a sound and clear analysis. If the MTT estimates are to be kept, I think they need to be made much more robust through uncertainty analysis and/or mechanistic model explorations (the GLUE presented just gets at modeling fitting). Further, the role of variability versus errors given limited sampling in time and space must be extensively considered. Given the audience of HESS I feel this loose application of MTT

convolution efforts weakens the case for this research and is not needed given there are several strong aspects already.

(2) Manuscript structuring

The introduction lacks any logical structure and must be improved. As currently presented, several topics are touch in an apparently random order. First montane landscapes then isotopes then MTT then tracers then EMMA and finally Kenya. The must be a general building of argument to highlight a knowledge gap that this study is trying to fill. The review of literature is rather surficial and must be improved to highlight better the current landscape surround this study to help the reader see where this study fits in with previous work.

In addition, there is some concern with regards to mixing results in with methods. The section 2.5.2 is a good example of this. To alleviate this, I would recommend adding a results section whereby you present the raw data collected (isotopic and chemistry) and characterize these data fully. That type of an overview and statistic presentation will then lay a groundwork for the more advance results. In practice, this means to expand sections 3.1 and 3.2 and allow the data to take center stage for this study – which is valid as these data are a significant contribution to the literature. As such, the data collected should be thoroughly reviewed and presented for the reader.

Last, the discussion section left a bit to be desired. I felt there was much text in this section that could find a better home in results as it just highlights the findings of this current study. There could be expansion on the limitations and implications of this study for the region or these types of regions. That shift in emphasis would likely resonate better with readers helping this study move from a place-based investigation to a more general research investigation.

---

## Referee Comment (RC3) · Anonymous Referee #3 · 19 Mar 2018

The study of Jacobs et al. used weekly data of the isotopic and chemical composition of streamwater, precipitation and other end members in three nested catchments with different land use in Kenya, Africa, to analyze how differences in land use may affect streamflow generation. To test this, the authors used end-member mixing analysis to estimate the relative contributions of the end-members to streamflow, as well as use a convolution approach to calculate the mean transit times of streamwater at catchments with different land use. While I consider the data set and the research question relevant for the readers of HESS, there are some parts that should be addressed before publication.

[Figure]

The mean transit time (MTT) estimates based on a data set covering ca. 1.5 years are likely to be highly uncertain. This is evident, for instance, in the similar numbers of NSE, RMSE and Bias for the streamwater and soil water samples at the sites SHA and TF (Table 3): While streamwater was sampled weekly at these sites (n>100), MTT estimates were similarly uncertain for streamwater as for soil water - from which only a small number of samples was collected (n<17)! Thus, based on the model performance criteria presented in the manuscript, I would not strictly believe the values obtained for streamwater either. Although the authors elaborate on the shortcomings of their data set with regard to estimate MTTs (Sect. 4.3), they do not consider using an alternative approach such as the young-water fraction framework (Kirchner, 2016a, b). This framework uses the seasonal cycle amplitudes of streamwater and precipitation amplitudes to estimate the fraction of water younger than ca. 3 months. Thus, with the data set presented by Jacobs et al., such an analysis might result in estimates of the young-water fractions of streamwater that are more robust than the MTTs. (Using the soil water samples from the sites NF and OUT might also reveal some interesting results, however, the data from the sites SHA and TTP are clearly too incomplete for such an analysis.)

In the catchment SHA, the samples from a wetland (WL, n=4) and the shallow well (WE.b, n=2) comprised two important end-members in the 3-component mixing analysis, whereas no wetlands or shallow wells were sampled in the other two catchments. Thus, I question the comparison made between the three sub-catchments: the relative contribution of precipitation at a site is inevitably linked to the contributions of the other two end members (all components must add up to 1), and therefore the precipitation contributions of NF and TTP cannot simply be compared with the precipitation contributions of SHA.

In general, I find the presentation of the solute concentrations of the different end members and streamwater insufficient - although this data set builds the foundation for the whole study. In the box plot (Figure 2) it is very difficult to distinguish between the different sites (vertical gridlines would help here) and end-members (distinction between the different end members would be impossible in a BW print). I suggest that the authors elaborate more on the data set, incl. uncertainties and times of sampling. Are the times of sampling representative for the flow regime at the sites or were the samples only collected during low-flow conditions? A presentation of the data similar to Figure 4 might be useful for this.

Abstract:

- The numbers presented in p1, L27-29 for the average relative contributions of springs and wetlands to streamwater are confusing: wetlands were only analyzed for one catchment (SHA), and in the Abstract it appears as if wetlands and springs were considered equivalent end members. In addition, I don't understand how the numbers presented in p1, L29-31 confirm that "... catchment hydrology is strongly influenced by land use, which could have serious consequences for water-related ecosystem services, such as provision of clean water.". Do the authors compare agricultural (i.e., de-forested) catchments to an un-altered forested catchment (i.e., baseline scenario)? If this is the case, then the results should be presented within such a framework.

Introduction:

- The different sub-sections of the introduction should be linked better. For instance, paragraphs 1 and 2 present two very different topics (tropical montane catchments and stable water isotopes, respectively), which have to be put into a common context, otherwise the reader is lost.

- The authors hypothesize that (a) streamwater in the natural forest sites is (on average) older than streamwater in agricultural catchments (smallholder agriculture, tee and tree plantations); (b) precipitation comprises a larger fraction of streamflow in the agricultural catchments than in the naturally forested catchment; and (c) that seasonality in rainfall causes temporal variability of these streamwater sources throughout the year. The formulations of the working hypotheses (a) and (b) are somewhat redundant: when streamflow at site A contains more precipitation (i.e., "new" water) relative to another site B, we should expect the mean transit time of Site A to be shorter. Thus, hypothesis (a) results from hypothesis (b). Regarding hypothesis (c), I don't understand how accepting/rejecting this statement adds to the conclusions of this study. The authors discuss hypothesis (c) only briefly later in the manuscript (p11, L21-23), which makes me wonder why it is stated so prominently in the Introduction?

Methods:

- P3,L30: What are the areal fractions of different land-use types in the main catchment (OUT)? This information would also be required to elaborate on the authors' statement on p13, L18-20: "One could also expect that, since OUT is a mixture of the three land use types dominating the sub-catchments, the MTT should be similar to or an average of the estimated MTTs of the sub-catchments.". This statement would only be true if the three sub-catchments are representative for the areal fractions of land use in the main catchment. - 2.3 Sampling and laboratory analysis: What are the instruments' measurement precision and accuracy? Especially in the case of Li, the measured concentrations (« 1ug/L) might be highly uncertain for precipitation and throughfall. Results: - 3.2 Isotopic composition: "There was no significant effect of elevation on $\delta$18O values of the precipitation samples, but precipitation samples collected at higher altitude (SHA-PC) were generally more depleted than those collected at lower altitudes (NF-PC, TTP-PC and OUT-PC).". This sentence is confusing, please reformulate.

- Figure 6 and analysis of Figure 6: Some of the relative contributions are highly uncertain, however, I miss a proper uncertainty analysis here. Although the authors discuss various sources of uncertainty in Sect. 4.2., a quantitative uncertainty analysis is still missing. At least, showing the error bars in Figure 6 would be helpful to interpret the results with more caution (i.e., Could the variability of the end members be an artefact of uncertainty in the EMMA?, p11 L21-23)). In addition, the Abstract, the authors present the average contributions without any uncertainty measures, which might be misleading. Discussion: - 4.2 Dominant water sources: Based on another study in the

NF catchment (Jacobs et al., in review) the authors conclude that in the NF catchment precipitation reaches the stream network via shallow sub-surface flow. Short residence times in the shallow subsurface thus result in dilution effects in streamwater. However, for the TTP catchment, the authors claim that "...surface runoff could have a different chemical signature than precipitation..." (p11, L13), which somewhat contradicts their previous statement in L3: "Therefore, if event water, i.e. precipitation or throughfall, is only in contact with the soil for a short time (e.g. several hours), the chemical composition of the water that enters the stream might be comparable to the composition of precipitation or throughfall.". Please clarify this.

Minor comments: P8, L13: Where these evaporated samples used in the analysis? Please clarify. P9, L28: "... has been observed elsewhere as well." – Where exactly? Are these sites comparable to the sites of this study? P12, L15: An alternative method to sample soil water would be suction lysimeters. P14, L23: "Due to the similar soils..."

References: Kirchner, J. W.: Aggregation in environmental systems-Part 1: Seasonal tracer cycles quantify young water fractions, but not mean transit times, in spatially heterogeneous catchments, Hydrol. Earth Syst. Sci., 20, 279-297, 10.5194/hess-20-279-2016, 2016a.

Kirchner, J. W.: Aggregation in environmental systems-Part 2: Catchment mean transit times and young water fractions under hydrologic nonstationarity, Hydrol. Earth Syst. Sci., 20, 299-328, 10.5194/hess-20-299-2016, 2016b.

---

## Referee Comment (RC4) · Anonymous Referee #4 · 20 Mar 2018

This study investigates the effects of land cover on water sources and flow paths in a montane catchment in East Africa. This case study could be important in filling in the gap of hydrologic knowledge in an understudy landscape. However, I have serious concerns about the suitability of the collected data to complete the analysis conducted by the authors and to provide evidence to answer the research questions. In addition, the manuscript is vague and incomplete. I believe the manuscript is not ready for publication. Below are some specific comments: 1) Given that, the available isotopic data is only 1.5 years long the authors should provide an assessment of the uncertainty in the computed MTT? The performance of the fits by the Gamma and EPM are actually similar yet the MTT for OUT_S15 was different between these two models. How do

you explain that? It is not clear how the authors chose the Gamma and EPM functions. Did they consider what model had better constrained parameters? In addition to the modelling shortcomings, how can MTT estimates calculated from 1.5 years of data provide information about the hydrologic impacts of different land covers? I wonder if a first step should be a hydrometric analysis that compares land covers and that can informed the findings form the MTT in light of physical processes. In addition, there might be interesting patterns in the isotopic data alone in terms of means per location, per season, comparisons across soil, stream, groundwater, and precipitation that would allow contrasting the different land covers. I am looking a figure 3 thinking: there is many data that have not been appropriately described in the paper. My point is that the isotopic data can we used on other ways different from in convolution equation for MTT. 2) The organization of the paper and its content is insufficient. a. The introduction is no short and does not set up the problem well. It is not clear what would the contribution of this study be nor how it fits with previous literature. b. Methods: It to short and refers the reader to a paper in review. A more comprehensive description is in order. The methods indicated that precipitation was estimated using Thissen polygons based on the information (I assumed, from the nine tipping buckets) however the results from this analysis is never presented in the results section. How variable is precipitation in space and time in this system? c. The result section is vague. For instance on 3.1. (Solute concentrations) the authors do not describe any one solute but instead talk all simultaneously as high or low. The result sections should include some actual numbers so that the reader knows what low or high mean. Likewise, there is no information in the results about how the values for the isotopic concertation vary in space and time per precipitation, stream, soil water, etc.

---

## Author Comment (AC1) · 4 Apr 2018

We would like to thank Reviewer #1 for his valuable feedback on our manuscript. The reviewer identified several major issues with the methodology used. Here we would like to reply to the major comments made by Reviewer #1:

1) The MTT methodological explanation is adequate (if some citation of GLUE development papers is included) but it fails to describe how a (400?) year-long 18O input function has been obtained to feed the lumped models when the rainfall sampling period was just 75 weeks long.

Reply: Bracketed values shown in the third column of table 2, i.e., tau = [1-400], correspond to the range of values that the MTT parameter could take for solving the convolution integral. By mistake, the units for these values are missing. This will be corrected during the revision of the manuscript. The equations were fed with weekly data, which was the interval of the sampling campaign. Therefore, the value of 400, means 400 weeks (=7.7 yr), which is a long enough period to cover the maximum possible values that the MTT could take for solving the convolution function. According to literature, it is appropriate to use stables isotopes of water for MTT estimations of up to 4 or 5 years. Regarding to the 'limited' length of sampling period (75 weeks for the isotopic signal of rainfall), used to feed the lumped models, we hypothesize a constant interannual recharge of the aquifers. We acknowledge that, ideally, it is advisable that the length of the sampled period is at least comparable to (or longer than) the length of the estimated MTT. However, for remote tropical montane catchments, data is generally scarce because of limited funding, harsh meteorological conditions and challenging accessibility. On the other hand, an advantage of tropical areas compared to temperate zones, is the low interannual and intra-annual variability in terms of meteorological characteristics like temperature and precipitation. Based upon this reasoning, preliminary insights of the rainfall-runoff characteristics of a tropical catchment could be derived from assuming that the seasonality of the isotopic signal of one year could resemble that of another. In this sense, it is reasonable to artificially extend a short input time series (rainfall) through repeating the available sampled time series in a loop. For our case, the input isotope time series were repeated 20 times. Repeating the input time series in a loop is a common practice where input data is limited, and not only for studies of tropical montane forest (Muñoz-Villers and McDonnell, 2012; Timbe et al., 2014; Hrachowitz et al., 2010 and 2011).

2) It has been shown that MTT determinations using seasonal variations of tracer signals (such as the 18O one) cannot provide acceptable results longer than a few months in stream (mixed) waters due to the strong non-linearity of the driving function (Kirchner, 2016).

Reply: The approach presented by Kirchner (2016) is a valuable contribution to the study of the rainfall-runoff behavior of natural heterogeneous systems, which, by the way, need to be studied assuming non-linearity. As pointed out in the referred publication, the estimation of MTT through tracer cycles and methods like the lumped convolution approach, as used in our manuscript, should be limited to homogeneous catchments for which steady state conditions apply. We acknowledge that natural systems are implicitly heterogeneous, however their degree of heterogeneity could be highly variable. How heterogeneous or homogenous does a catchment have to be to prevent, or allow, the use of traditional approaches like lumped parameter models? Some ideas are posed in the referred work: Figure 4 in Kirchner (2016) shows two contrasting outflow isotope signals: a highly damped signal and another whose amplitude closely resembles the one of rainfall. This means that before using the traditional approach in the study of nested catchments, we should first check if the amplitudes of the sampled sites are comparable. For instance, if two elements (sub-catchments) of a nested catchment have contrasting amplitudes, then we should be aware that the combination of these two outflows will provide an unrealistic amplitude and therefore an unrealistic MTT. On the other hand, if amplitudes of isotope signals of outflows are similar, it will be a preliminary indication of homogeneity (i.e., little to moderate heterogeneity), and therefore traditional steady state approaches could be applied. The standard deviation ($\sigma$) is a proxy of the amplitude of the isotopic signal (e.g., Garvelmann et al., 2017). For our data, $\sigma$ values for the observed input functions (i.e., rainfall) for every catchment are 2.59‰ 2.73‰ and 2.54‰ for NF, SHA and TTP, respectively. On the other hand, $\sigma$ for outflows (for the same period, according to the observed data) are 0.10‰ 0.11‰ and 0.07‰ (for NF-RV, SHA-RV, and TTP-RV, respectively). Then, an intercomparison between the amplitudes of every river outflow can be easily performed through a simple fourth proportional calculation: 12.4%, 11,4% and 10.3%, (for NF, SHA and TTP, respectively). The latter values correspond to the amplitude of the outflows compared to the original amplitude of the rainfall, expressed in percentages. The similarity between the amplitudes of the three analyzed catchments is an indicative of homogeneous characteristics, meaning that they could be characterized by a single transit-time distribution. Furthermore, the young water fractions (YWF) of NF-RV, SHA-RV and TTP-RV, calculated as a ratio between the amplitude of the outflow (stream water) and input signal (rainfall) are also similar: 3.72%, 4.08%, and 2.67%. These little portions of YWF, show that analyzed stream waters correspond to baseflow dominated catchments in which steady state conditions could apply. In this respect we should have in mind that due to the highly damped signal of the analyzed outflows, no discrimination was performed between samples taken during baseflow or high-flow conditions (i.e. all 75 stream water samples for each catchment were included in the analysis). Another way to check if homogeneity was correctly assumed could be to check if just one type of transit time distribution function provides the best results for all the analyzed sub-catchments and/or the best parameters of that single function are similar or comparable among catchments. For our case, the gamma model provided the best fitting efficiencies, and the model parameters were also similar for two out of the three analyzed catchments (the results for TTP-RV were discarded because of its low fitting efficiency, NSE=0.05).

3) For such damped tracer signals in the stream waters and low model efficiencies, much larger MTT uncertainties should be obtained, showing results coherent with point 2. My opinion is that the small uncertainties obtained are an artefact due to the way the behavioral models have been selected in the GLUE exercise. Accepting only parameter sets with efficiency just 5% lower than the optimal one might be appropriate for high efficiency values, but not in the case of such low efficiency values because the range of behavioral parameters becomes too narrow. Some GLUE published works dealing with large uncertainties sensibly used all parameter sets with positive efficiencies. Alternatively, all the parameter sets with such low efficiencies might be rejected as a way to resolve that the method is inappropriate.

Reply: We agree with the comment of the reviewer 1: if the adjustments of the model are low, a range of 5% below the best solution, becomes narrow. For all our analyzed stream waters, the model efficiencies were low, mainly due to the highly damped isotopic signal (see point 2). The calculated uncertainties are just a way of comparing the degree of identifiability of the model parameters, among the sub-catchments under study. These results (i.e., ranges of solutions) are not meant for comparison with ranges of uncertainty of other studies with substantially higher NSE. However, since scatterplots of behavioral solutions are not presented in the paper, the presented ranges are useful for the reader to know if the models converge to a unique best solution or not (i.e. whether behavioral solutions tend to a peak or if they have a flat shape). In the section 4.3 of our paper, we acknowledge that the fitting efficiency was too low for TTP-RV (NSE = 0.05) and therefore their associated results should not be considered. In the same section we also acknowledge that for these cases: "Better predictions could be obtained by using more appropriate tracers for estimating transit times of several years to decades, such as tritium (3H) (Cartwright et al., 2017). A longer sampling period of at least 4 years would also improve the reliability of the mean transit time estimates (McGuire and McDonnell, 2006)..." In this respect, in the revised version of the paper we will emphasize that due low fitting efficiencies of stream waters, results for SHA-RV and NF-RV should also be taken with care, and only as an indicative that MTT of several-years older. In the case of stream waters, I suggest to remove the proposed MTT determinations, unless the above points are adequately answered. The authors may reasonably continue using the clear damping of the tracer signal in the stream waters as an indicator of several-year old waters, and even the differences in the temporal variability of the tracer signals might be used to indirectly rank the waters MTTs. In the case of soil mobile waters, I suggest the application of some analysis of the significance of MTT differences found, using the MTTs likelihood distributions provided by the GLUE exercise. As requested by Reviewer 1, after clarifying some aspects (detailed previously in this reply), which will be explicitly included in the revised version of the paper, we believe that it will be important to keep the MTT determinations since these preliminary results will provide a base knowledge in this remote study area for which no previous data was available. We welcome the suggestion

of Reviewer 1 for soil mobile waters: to include a sort of MTTs likelihood distribution function analysis. In this respect, cumulative density functions (CDF) (together with their respective analysis and discussion) for the three analyzed soil water sites and for both models: EPM and GM, considering the range of associated uncertainty, will be included in the revised version of the paper.

4) The third but also relevant issue refers to the End Member Mixing Analysis (EMMA) for the Small Holder Agriculture (SHA) stream waters. The use of the well SHA-WE.b as end member representative of groundwater chemistry is not reasonable. One well in the headwaters with solute concentrations very different from those in other nine wells may represent either a different water source or some pollution effect, but it is not sensible to hypothesize that it can be a relevant source for stream water when its chemistry is very local as it is not transmitted to the other well waters. If well understood, the use this end member with very low contributions as representative of groundwater is depicted in Figure 7 (b), although this is inconsistent with some text in the conclusions: "A second, different groundwater source was identified in the smallholder agriculture catchment, which was an important end member during baseflow"

Reply: We understand the concern of the reviewer about the use of a single 'outlier' to explain stream water chemistry and mixing of different end members in a catchment and, ideally, we would have identified another end member which would fit the end member model better than SHA-WE.b. Nevertheless, of the sampled end members, SHA-WE.b is the only end member that can explain the stream water chemistry of samples taken during the dry season (Figure 5 in manuscript). As mentioned in the discussion (P. 11 L. 29-31), it is likely that end members are missing and a further effort should be made to develop more appropriate and less uncertain end member mixing models. Considering that this is the first effort to characterize hydrological flow paths and water provenance in this tropical montane area, we think that the use of SHA-WE.b is reasonable to present preliminary findings, given the available data. With regard to the question whether SHA-WE.b is a polluted well or represents a different groundwater source, the chemical composition (high concentrations of Si, Li, K, Na and Rb) of SHA-WE.b resemble the elements that could be expected in groundwater based on the geology of the area. We consider both the wetland SHA-WL and SHA-WE.a/SHA-WE.b groundwater sources. Since SHA-WL is an important contributing groundwater end member, especially during the wet seasons, referring to SHA-WE.b in the conclusion as a second groundwater source seems reasonable. However, we indeed need to revise the sentence referred to by the reviewer, as the use of the word 'important' overestimates the potential role of this end member in stream flow generation.

Aside from these major issues, other points indicated by Reviewer #1, such as the title of the manuscript, the use of the expression 'soil water' and the detailed comments will be addressed in the revised version of the manuscript.

References: Garvelmann, J., Warscher, M., Leonhardt, G., Franz, H., Lotz, A. and Kunstmann, H.: Quantification and characterization of the dynamics of spring and stream water systems in the Berchtesgaden Alps with a long-term stable isotope dataset, Environ. Earth Sci., 76(22), 766, doi:10.1007/s12665-017-7107-6, 2017. Hrachowitz, M., Soulsby, C., Tetzlaff, D., Malcolm, I. A. and Schoups, G.: Gamma distribution models for transit time estimation in catchments: Physical interpretation of parameters and implications for time-variant transit time assessment, Water Resour. Res., 46(10), W10536, doi:10.1029/2010WR009148, 2010. Hrachowitz, M., Soulsby, C., Tetzlaff, D. and Malcolm, I. A.: Sensitivity of mean transit time estimates to model conditioning and data availabilityle, Hydrol. Process., 25(6), 980–990, doi:10.1002/hyp.7922, 2011. Kirchner, J. W.: Aggregation in environmental systems – Part 1: Seasonal tracer cycles quantify young water fractions, but not mean transit times, in spatially heterogeneous catchments, Hydrol Earth Syst Sci, 20(1), 279–297, doi:10.5194/hess-20-279-2016, 2016. Muñoz-Villers, L. and McDonnell, J.: Runoff generation in a steep, tropical montane cloud forest catchment on permeable volcanic substrate, Water Resour. Res., 48(9), W09528, doi:10.1029/2011WR011316, 2012. Timbe, E., Windhorst, D., Crespo, P., Frede, H.-G., Feyen, J. and Breuer, L.: Understanding uncertainties when inferring

mean transit times of water trough tracer-based lumped-parameter models in Andean tropical montane cloud forest catchments, Hydrol Earth Syst Sci, 18(4), 1503–1523, doi:10.5194/hess-18-1503-2014, 2014.

---

## Author Comment (AC2) · 8 Apr 2018

We would like to respond in detail to point 1 (Data presentation and analysis) provided by the reviewer. With regards to point 2 (Manuscript structuring), we are grateful for the suggestions provided by the reviewer and will incorporate the feedback (improved structuring and content of introduction and discussion, as well as presentation of the raw data collected) into the revised version of the manuscript.

1) There is some concern with taking these isotopic data into the convolution modeling. Specifically, the length of observation record is not overly long which limits the ability to map out some realistic travel times here. The uncertainty gets very high in this regard.

[Figure]

So, at best one could argue that the MTT estimations are just first-order assessments for comparing between the catchments. Further, taking on a classic time invariant MTT estimation is a bit troublesome with regard to the potential for comparisons. Namely, the travel times for each catchment likely shift with wetness (storage) condition and this dynamic shifting (mixing) likely represents itself differently throughout the one and a half year been considered here. As such, it is difficult to separate the impact of the land use on MTT from the variability of the flow on the MTT – more likely these aspects compound (and confound) the issue (see van der Velde et al., 2015, Consequences of mixing assumptions for time-variable travel time distributions, Hydrological Processes).

Reply: We acknowledge that the application of the convolution method to estimate MTTs, generally applied for time-invariant conditions, does not allow a precise estimation of the effect of land use on MTT. The main reason is that the high damping of the stream water signal compared to the amplitude of the isotope signal of rainfall, indicates high MTTs of water. In this case, where the MTT is of an order of magnitude of years, it could be difficult to identify subtle differences in the order of days, weeks or a few months caused by differences in land use. Furthermore, as it has been stated by Velde et al. (2015), it is also possible that changes in MTT occur on an inter-annual basis, depending on the soil's previous moisture conditions. We argue, however, that the three sub-catchments are subject to similar rainfall patterns, as can be seen in the weekly precipitation time series presented in Figure 4 of the manuscript. Furthermore, differences in storage and flow and resulting differences in MTT can from our view be attributed to a land use effect as geological differences between the three sub-catchments are minimal. We do, however, agree with Reviewer #2 that the MTTs presented here are unlikely to be the 'true' MTTs of the catchments. Due to the limited information on transit times of tropical montane catchments of Africa, compared to the amount of information available for mountainous catchments of high latitudes, we consider that it is necessary to include the present stable isotope data and their respective MTT estimations in the manuscript. In this regard, preliminary but relevant information, as expressed by Reviewer #2, is the knowledge of at least the order of

magnitude of MTTs, which, for stationary conditions, can be obtained by the convolution method. Although the present outcomes are preliminary findings, they could serve as the baseline for future studies in which a greater number of samples could be used or a longer sampling period, which will allow the use of more sophisticated methods (e.g. time-variant approaches) and thereby more subtle differences in the movement of soil water could be accounted. In a revised version of our manuscript, we will emphasize that the present isotope data and the respective estimates derived from them, serve a preliminary characterization of the water MTTs in the analysed catchments, for stationary conditions. Based on the magnitude of the estimated MTTs of stream water (of the order of years), for stationary conditions, we will also emphasize that with the current available data it is not possible to discern whether these small differences (among catchments MTTs) are actually caused by differences in land use. In addition, regarding the assumption of stationary conditions for the present work, the criteria for which this assumption was adopted will be explained in detail, and, due to the limited sampling period, the scope of the present MTT estimates will be emphasized accordingly. On the other hand, a more detailed analysis of the observed data will also be performed: the amplitude of the observed input and output isotope signals will be contrasted, including the estimation of the young water fraction (YWF) (Kirchner, 2017) (see reply to Reviewer #1). With respect to the uncertainty associated to MTT of stream water, we will include, in a supplementary material, plots of the best fitting efficiencies and the associated GLUE uncertainties. In the same way, we will focus on a more detailed discussion of these aspects.

---

## Author Comment (AC3) · 8 Apr 2018

We would like to thank Reviewer #3 for the feedback provided and will respond to the major comments provided by the reviewer. Other suggestions, e.g. presentation of the raw data, clarification of the abstract, improvement of structure of introduction, revision of the hypothesis, will be incorporated in the revised version of the manuscript.

1) The mean transit time (MTT) estimates based on a data set covering ca. 1.5 years are likely to be highly uncertain.

Reply: With respect to the concern of Reviewer #3, about the limited sampling period

(1.5 y), in a revised version of our paper we will point out that present MTT estimates for stream water, are meant for a general characterization and comparison of the travel times of water between the studied sub-catchments, for stationary conditions. We will also emphasize that although preliminary, this information will serve as baseline for future research in which methods like time-variant techniques, could be used. We will emphasize that such approaches could serve as better tools to study the associated effects of land use on travel times or water flow paths.

2) This is evident, for instance, in the similar numbers of NSE, RMSE and Bias for the streamwater and soil water samples at the sites SHA and TF (Table 3): While streamwater was sampled weekly at these sites (n>100), MTT estimates were similarly uncertain for streamwater as for soil water - from which only a small number of samples was collected (n<17)! Thus, based on the model performance criteria presented in the manuscript, I would not strictly believe the values obtained for streamwater either.

Reply: We would like to clarify that the goodness of fit (NSE), RMSE and Bias, are not the same for soil water and stream water. Since the observed isotope signals of soil water have a larger amplitude than of stream water, estimations of MTT of soil waters have, in general, higher NSE, but also higher values of RMSE and Bias. Further, neither the associated uncertainties of estimations of MTT for soil water are comparable to those of stream water: uncertainty ranges for stream water (Table 3) are expressed in years while for soil waters are of the order of weeks (Table 4). Regarding the number of observation samples taken for the convolution approach in order to estimate MTTs, for stream water we took n = 75 samples for each of the four evaluated catchments (not n> 100 as being suggested by Reviewer #3), while for MTT estimations of soil water sites, the number of samples was n = 47 (not n <17 as it is stated by the Reviewer #3). For soil water sites, a larger number of samples was not possible since there were weeks in which the mobile soil water collected by the wick samplers was insufficient or non-existent.

3) Although the authors elaborate on the shortcomings of their data set with regard to

estimate MTTs (Sect. 4.3), they do not consider using an alternative approach such as the young-water fraction framework (Kirchner, 2016a, b). This framework uses the seasonal cycle amplitudes of streamwater and precipitation amplitudes to estimate the fraction of water younger than ca. 3 months. Thus, with the data set presented by Jacobs et al., such an analysis might result in estimates of the young-water fractions of streamwater that are more robust than the MTTs. (Using the soil water samples from the sites NF and OUT might also reveal some interesting results, however, the data from the sites SHA and TTP are clearly too incomplete for such an analysis.)

Reply: In a revised version of our paper we will include estimations and the respective analysis of the Young Water Fractions of streamwaters (YWF). The base of our analysis will be the amplitudes of the observed isotopic input and output signals, and according to the criteria established by Kirchner (2017). A preliminary estimate has been already posted as part of the reply to comments of the Reviewer # 1.

4) In the catchment SHA, the samples from a wetland (WL, n=4) and the shallow well (WE.b, n=2) comprised two important end-members in the 3-component mixing analysis, whereas no wetlands or shallow wells were sampled in the other two catchments. Thus, I question the comparison made between the three sub-catchments [. . .].

Reply: The main reason that no shallow wells or wetlands were sampled in the two other sub-catchments, was that there were no similar accessible wetlands in these two sub-catchments. Specifically the natural forest sub-catchment is highly inaccessible due to the dense vegetation and absence of footpaths. We assume, however, that springs and wetlands represent similar groundwater sources, as supported by their similar chemical composition (P. 10, L. 3-4) and are therefore comparable between the three sub-catchments. This hopefully also clarifies the reviewer's comment: '[. . .] wetlands were only analyzed for one catchment (SHA), and in the Abstract it appears as if wetlands and springs were considered equivalent end members'. With regards to shallow wells, as mentioned in the discussion (P. 12, L. 8-9) there are no shallow wells in the forest, because of absence of habitation, and within the tea plantations

shallow wells are not present as all settlements have access to piped water. Therefore, although we tried to include similar end members for each sub-catchment in the design of the study, we were limited in the availability and accessibility of end member sampling sites. We acknowledge in the discussion that it is likely that end members are missing due to e.g. lack of proper groundwater access and that the results have therefore a quite large uncertainty.

---

## Author Comment (AC4) · 8 Apr 2018

We would like to thank Reviewer #4 for the feedback and will respond to the major comments provided by the reviewer. Other suggestions will be incorporated in the revised version of the manuscript.

1) Given that, the available isotopic data is only 1.5 years long the authors should provide an assessment of the uncertainty in the computed MTT? The performance of the fits by the Gamma and EPM are actually similar yet the MTT for OUT_S15 was different between these two models. How do you explain that? It is not clear how the authors chose the Gamma and EPM functions. Did they consider what model

had better constrained parameters? In addition to the modelling shortcomings, how can MTT estimates calculated from 1.5 years of data provide information about the hydrologic impacts of different land covers?

Reply: A revised version of the manuscript will include detailed plots of the best solutions and the associated uncertainty ranges, for every stream and soil water site as supplementary information. Indeed, the performance of the fits to the objective function (NSE) for Gamma and EPM are quite similar, however it is not true that the related MTT estimations differ, we acknowledge this in the last paragraph of the Discussion section when referring to soil water sites (P. 14, L. 1–7). In this regard, if we use a parameter of $\eta$=1 when using EPM, or $\alpha$=1 when using GM, then GM or EPM becomes a simpler EM, whose only parameter is MTT. For the case of OUT_S15, the MTT estimation (using GM or EPM, with $\alpha$ or $\eta$ = 1, respectively) is 7.24 weeks. For stream water sites (except for the case of TTP-RV, which results were not considered for analysis due to the low NSE) it was easy to choose the best performing model: for NF-RV, SHA-RV and OUT-RV, according to NSE, the best performing model was GM (Table 3). Furthermore, the justification of model selection is described in Section 2.5.1 (P. 6, L. 29-32): 'Among the diverse model types, two-parameter models such as the gamma model (GM) or 30 the exponential piston flow model (EPM) are commonly used for MTT estimations (Hrachowitz et al., 2010; McGuire and McDonnell, 2006) and were identified by Timbe et al. (2014) as most suited to infer MTT estimations of spring, stream and soil water in an Andean tropical montane forest catchment.' We agree that 1.5 years of data is not so much and ideally a the data input should cover a period as long as or longer than the MTT, but one has to consider that very little, and in this case no data was available for the study region. Furthermore, due to limited funding and accessibility in such remote areas, it become challenging to collect a long-term dataset for stable isotopes. Considering the conditions in the study area and the requirements for MTT analysis, we think it is reasonable to present the estimated MTTs for the three sub-catchments and main catchments as preliminary findings, as long as its uncertainty is emphasized.

2) I wonder if a first step should be a hydrometric analysis that compares land covers and that can informed the findings form the MTT in light of physical processes. In addition, there might be interesting patterns in the isotopic data alone in terms of means per location, per season, comparisons across soil, stream, groundwater, and precipitation that would allow contrasting the different land covers. I am looking a figure 3 thinking: there is many data that have not been appropriately described in the paper. My point is that the isotopic data can we used on other ways different from in convolution equation for MTT.

Reply: As suggested by other reviewers, we will expand the presentation of the raw isotope data to give it a more prominent position in the manuscript. This will hopefully also address the concern of Reviewer #4 that not all data has been described appropriately in the paper. While presenting the isotope data in more detail we will include the calculation and analysis of the Young Water Fraction (YWF) (Kirchner 2017) of the analyzed catchments.

3) The organization of the paper and its content is insufficient. a. The introduction is no short and does not set up the problem well. It is not clear what would the contribution of this study be nor how it fits with previous literature.

Reply: This will be addressed in the revised version of the manuscript.

4) Methods: It to short and refers the reader to a paper in review. A more comprehensive description is in order. The methods indicated that precipitation was estimated using Thiessen polygons based on the information (I assumed, from the nine tipping buckets) however the results from this analysis is never presented in the results section. How variable is precipitation in space and time in this system?

Reply: The current version of the manuscript is already quite long. The study area and collection of discharge and precipitation has been described extensively in other publications (e.g. Jacobs et al. 2017 and the manuscript under review, which is now published as Jacobs et al. 2018). We therefore decided not to repeat this in the current

manuscript. The precipitation results have indeed not been presented explicitly in the results, but differences in rainfall between the four catchments are displayed in Figure 4 as weekly precipitation. This also clearly shows the temporal variation in precipitation. We will consider presenting some of the information you request in a supplement, as we do not think that such detailed information is relevant for the manuscript without making it too long.

5) The result section is vague. For instance on 3.1. (Solute concentrations) the authors do not describe any one solute but instead talk all simultaneously as high or low. The result sections should include some actual numbers so that the reader knows what low or high mean. Likewise, there is no information in the results about how the values for the isotopic concertation vary in space and time per precipitation, stream, soil water, etc.

Reply: This will be addressed in the revised version of the manuscript. Similar to the precipitation data, more detailed information will be presented in a supplement.

References:

Hrachowitz, M., Soulsby, C., Tetzlaff, D., Malcolm, I. A. and Schoups, G.: Gamma distribution models for transit time estimation in catchments: Physical interpretation of parameters and implications for time-variant transit time assessment, Water Resour. Res., 46(10), W10536, doi:10.1029/2010WR009148, 2010.

Jacobs, S. R., Weeser, B., Guzha, A. C., Rufino, M. C., Butterbach-Bahl, K., Windhorst, D. and Breuer, L.: Using high resolution data to assess land use impact on nitrate dynamics in East African tropical montane catchments, Water Resour. Res., 54, https://doi.org/10.1002/2017WR021592, 2018

Jacobs, S. R., Breuer, L., Butterbach-Bahl, K., Pelster, D. E. and Rufino, M. C.: Land use affects total dissolved nitrogen and nitrate concentrations in tropical montane streams in Kenya, Sci. Total Environ., 603–604, 519–532, 2017.

McGuire, K. J. and McDonnell, J. J.: A review and evaluation of catchment transit time modeling, J. Hydrol., 330(3-4), 543-563, doi:10.1016/j.jhydrol.2006.04.020, 2006.

Timbe, E., Windhorst, D., Crespo, P., Frede, H.-G., Feyen, J. and Breuer, L.: Understanding uncertainties when inferring mean transit times of water trough tracer-based lumped-parameter models in Andean tropical montane cloud forest catchments, Hydrol Earth Syst Sci, 18(4), 1503–1523, doi:10.5194/hess-18-1503-2014, 2014.
* * *

---

## Author Response (AR1)

Dear Editor, dear Referees,

We would like to thank you for the valuable feedback provided for our manuscript "Land use alters dominant water sources and flow paths in tropical montane catchments in East Africa" and for the opportunity to resubmit a revised version.

5 The comments from the referees were very helpful to improve the manuscript. We restructured the manuscript, included the Young Water Fraction approach (Kirchner, 2016) and re-evaluated our results, as suggested by the referees and the editor. Please find our point-by-point responses (in blue) to the comments of all referees (in black) below. Page and line number in the responses refer to the revised version of the manuscript (no tracked changes). We believe that the modifications based on the referees' comments have resulted in an improved manuscript and hope that it is now suitable for consideration for

10 publication as research paper in Hydrology and Earth System Sciences.

We look forward to hearing from you.

Kind regards,

15 Suzanne Jacobs

**Referee #1**

*General comments:*

The subject of the manuscript may be of interest for HESS readers, it represents a relevant work volume and is well presented,

20 but there are several formal and methodological issues that deserve a major revision of the manuscript before being acceptable for publication. The first issue is in the title of the paper itself. It is very assertive while the results of the work, taking into account the associated uncertainties, are much less convincing. So, I suggest to change the title or just to put it into an interrogative form.

Reply: We understand the concern of the referee and changed the title to: "Assessment of hydrological pathways in East

25 African montane catchments under different land use".

The second but main issue is respect to how Mean Transit Times (MTTs) have been obtained for stream waters. The several aspects of this issue are the following ones:

1) The MTT methodological explanation is adequate (if some citation of GLUE development papers is included) but it fails to

30 describe how a (400?) year-long $^{18}$O input function has been obtained to feed the lumped models when the rainfall sampling period was just 75 weeks long.

Reply: The values between brackets shown in the third column of Table 2, i.e., $\tau = [1–400]$, correspond to the range of values that the MTT parameter could take for solving the convolution integral. The units of this parameter (weeks) were unfortunately omitted from the original manuscript, but this has been corrected in the revised version. The 400 weeks (= 7.7 y) is a period

long enough to cover the maximum possible values that the MTT could take for solving the convolution function. According to the literature, it is appropriate to use stable water isotopes for MTT estimations of up to 4 or 5 years, but we mention the potential limitation of this method in P. 14, L. 28–30. We acknowledge that it is advisable the length of the sampling period to be at least comparable to (or longer than) the length of the estimated MTT. However, for remote tropical montane catchments, data are generally scarce because of limited funding and challenging accessibility. We added an explanation and a justification on how the data obtained during the sampling period were used to feed the lumped models (P. 7, L. 8–12).

2) It has been shown that MTT determinations using seasonal variations of tracer signals (such as the $^{18}O$ one) cannot provide acceptable results longer than a few months in stream (mixed) waters due to the strong non-linearity of the driving function (Kirchner, 2016).

Reply: We acknowledge that the approach presented by Kirchner (2016) is a valuable contribution to the study of the rainfall-runoff behaviour of natural systems and that natural systems are implicitly heterogeneous. We added P. 6, L. 20–31 to the methods to explain how we used the YWF as indicator of the degree of heterogeneity. The amplitude of precipitation and stream water signals and YWF in the three sub-catchments (presented in Section 3.4) suggest that the study area is fairly homogeneous, which would justify for the use of the convolution method for MTT estimates, as applied in our study.

3) For such damped tracer signals in the stream waters and low model efficiencies, much larger MTT uncertainties should be obtained, showing results coherent with point 2. My opinion is that the small uncertainties obtained are an artefact due to the way the behavioural models have been selected in the GLUE exercise. Accepting only parameter sets with efficiency just 5% lower than the optimal one might be appropriate for high efficiency values, but not in the case of such low efficiency values because the range of behavioural parameters becomes too narrow. Some GLUE published works dealing with large uncertainties sensibly used all parameter sets with positive efficiencies. Alternatively, all the parameter sets with such low efficiencies might be rejected as a way to resolve that the method is inappropriate.

Reply: We agree with the comment of the referee, and explained the choice and limitations of the applied subjective limits: "Due to the low fitting efficiencies and selected threshold of 5 % below the highest obtained NSE, the uncertainty bands for all sites were relatively narrow (Fig. S5–18). The uncertainties should therefore be considered as means of comparison of model parameters between sites and cannot be compared to uncertainties obtained in other studies with higher NSE values." (P. 10, L. 16–19). Additionally, we provided the results of the GLUE analysis as supplement to the manuscript (Fig. S5–18).

In the case of stream waters, I suggest to remove the proposed MTT determinations, unless the above points are adequately answered. The authors may reasonably continue using the clear damping of the tracer signal in the stream waters as an indicator of several-year old waters, and even the differences in the temporal variability of the tracer signals might be used to indirectly rank the waters MTTs. In the case of soil mobile waters, I suggest the application of some analysis of the significance of MTT differences found, using the MTTs likelihood distributions provided by the GLUE exercise.

Reply: After clarifying all aspects requested by the referee (detailed in responses to previous comments and incorporated in the revised manuscript), we believe that it is worth keeping the MTT determinations, since these results will provide a knowledge base in this remote study area for which no previous data are available. We included the cumulative density functions (CDF) for all analysed stream water and mobile soil water sites and for both models (EPM and GM), considering the range of associated uncertainty, as supplemental information to the manuscript (Fig. S19–20), and compared the transit time distributions of results from both models for each site.

The third but also relevant issue refers to the End Member Mixing Analysis (EMMA) for the Small Holder Agriculture (SHA) stream waters. The use of the well SHA-WE.b as end member representative of groundwater chemistry is not reasonable. One well in the headwaters with solute concentrations very different from those in other nine wells may represent either a different water source or some pollution effect, but it is not sensible to hypothesize that it can be a relevant source for stream water when its chemistry is very local as it is not transmitted to the other well waters. The analysis done can be shown as a test, but it cannot be taken as representative because groundwater contribution becomes underestimated and the other components overestimated. If well understood, the use this end member with very low contributions as representative of groundwater is depicted in Figure 7 (b), although this is inconsistent with some text in the conclusions: "A second, different groundwater source was identified in the smallholder agriculture catchment, which was an important end member during baseflow".

Reply: We understand the concern of the referee about the use of a single 'outlier' to explain stream water chemistry and mixing of different end members in a catchment and, ideally, we would have identified another end member which would fit the end member model better than SHA-WE.b. Nevertheless, of the sampled end members, SHA-WE.b is the only end member that explains the stream water chemistry of samples taken during the dry season (Fig. 4). As mentioned in the discussion (P. 13, L. 8–10), it is likely that end members are missing. Considering that this is the first effort to characterize hydrological flow paths and water provenance in Kenya, we think that the use of SHA-WE.b is reasonable to present preliminary findings, given the available data.

We argue that SHA-WE.b is a groundwater source based on its chemical composition: "Shallow well SHA-WE.b had trace element concentrations that were much higher than those of the other nine sampled shallow wells SHA-WE.a, but similar in magnitude to solute concentrations in a spring in the Andean Páramo (Correa et al., 2017) and deep groundwater in Tanzania (Koutsouris and Lyon, 2018). Since the trace elements with high concentrations in SHA-WE.b correspond with elements related to geology (e.g. Li, K, Na and Rb), it is likely that this source is groundwater-related." (P. 10–11, L. 31–4). Additionally, we consider both the wetland SHA-WL and SHA-WE.a/SHA-WE.b as groundwater sources. Since SHA-WL is an important groundwater end member, especially during the wet seasons, referring to SHA-WE.b as a second groundwater source seems a reasonable conclusion. We did, however, remove the questionable sentence from the manuscript.

Another more formal issue is the use of the 'soil water' expression to identify the samples of mobile waters sampled at different soil depths. In the current water isotope literature, 'soil water' refers to the total (bulk) water contained in the soil, including

mobile and immobile waters. In the methods section it is clearly justified that just mobile water was sampled, but in the abstract, figures and conclusions, some adjective such as 'mobile' or 'free' should be added to 'soil water' in order to avoid any misunderstanding.

Reply: We agree with the referee that this could cause misunderstanding and have therefore added 'mobile' to all references to the soil water samples throughout the manuscript.

*Specific comments:*

P. 3, L. 21: some hypothesis on how rain water reaches the stream should be added

Reply: We revised our hypotheses to: "Based on these results, we hypothesised that (a) the natural forest sub-catchment has a longer MTT than the tea plantation and the smallholder agriculture sub-catchments, because precipitation contributes less to streamflow in the forest catchment, and (b) the precipitation that contributes directly to streamflow will reach the stream through surface runoff in the tea plantation and smallholder agriculture sub-catchments and through shallow sub-surface flow in the forest sub-catchment." (P. 3, L. 20–24).

P. 7, L. 8: Nash & Sutcliffe (1970)

Reply: We included the reference in the sentence (P. 7, L. 19).

P. 8, L. 11: GLUE was first described in Beven & Binley (1992)

Reply: We included the reference in the sentence (P. 7, L. 22).

P. 3, L. 21: "Ten shallow wells (nine named SHA-WE.a and one SHA-WE.b)..."

Reply: We revised the sentence: "Ten shallow wells (nine named SHA-WE.a and one SHA-WE.b) in SHA were sampled twice." (P. 5, L. 24–25).

P. 8, L. 10 and 15: the units for the slopes are not correct.

Reply: Thank you, that is correct, we removed the incorrect units from the text (P. 8, L. 21; P. 8, L. 25).

P. 8, L. 15; P. 10, L. 9: this slope value seems too small looking to the graphs.

Reply: We repeated the analysis and had a closer look at the data. Although it does indeed look incorrect based on Fig. 2, the value for the slope is correct.

P. 8, L. 31: the contribution of precipitation to SHA stream waters is overestimated due to the role of SHA-WE.b commented above.

Reply: We agree with the reasoning of the referee, but decided to keep SHA-WE.b in the analysis for the reasons explained above. However, we emphasized that overestimation of the contribution of precipitation is likely due to the absence of a more appropriate end member: "However, the contribution of precipitation (57.4, 45.3–78.6 %) in SHA is probably overestimated due to the inclusion of shallow well SHA-WE.b as end member." (P. 12, L. 7–8).

**Referee #2**

*General comments:*

(1) Data presentation and analysis

There is some concern with taking these isotopic data into the convolution modeling. Specifically, the length of observation record is not overly long which limits the ability to map out some realistic travel times here. The uncertainty gets very high in this regard. So, at best one could argue that the MTT estimations are just first-order assessments for comparing between the catchments. Further, taking on a classic time invariant MTT estimation is a bit troublesome with regard to the potential for comparisons. Namely, the travel times for each catchment likely shift with wetness (storage) condition and this dynamic shifting (mixing) likely represents itself differently throughout the one and a half year been considered here. As such, it is difficult to separate the impact of the land use on MTT from the variability of the flow on the MTT – more likely these aspects compound (and confound) the issue (see van der Velde et al., 2015, Consequences of mixing assumptions for time-variable travel time distributions, Hydrological Processes). Some consideration of these aspect must be taken up within the analysis – or at a minimum in the discussion with regards to impacts on the results and interpretations. As these estimates are currently presented they tend to over-sell the ability of such analysis and what we can truly learn from them. These MTT estimation techniques are far from perfect and difficult to connect with mechanisms. It would be unfortunately for the uncertainty inherent in them to conflate with our understanding of these sites. Actually upon deeper reading, I am not sure the MTT analysis is truly justified or even needed in this study. The high uncertainty and extrapolation needed to make the convolution effort make sense and to interpret the results are not justified. This study would be more powerful to be a data presentation with an EMMA analysis constrained by the uncertainty inherent in these data which were hard to collect. The current MTT analysis is just too thinly supported by the minimal data and has no real consideration of variability versus uncertainty to allow for a rigorous interpretation. I would strongly recommend removing these parts of the study and focusing in on the other aspects to make for a sound and clear analysis. If the MTT estimates are to be kept, I think they need to be made much more robust through uncertainty analysis and/or mechanistic model explorations (the GLUE presented just gets at modeling fitting). Further, the role of variability versus errors given limited sampling in time and space must be extensively considered. Given the audience of HESS I feel this loose application of MTT convolution efforts weakens the case for this research and is not needed given there are several strong aspects already.

Reply: We thank the referee for the insights provided. We certainly acknowledge that the application of the convolution method to estimate MTTs, generally applied for time-invariant conditions, does not allow a precise estimation of the effect of land use on MTT. However, due to the limited information on transit times for tropical montane catchments of Africa, compared to the amount of information available for montane catchments of high latitudes, we consider valuable to include our stable isotope data and the MTT estimations in the manuscript. In the revised manuscript, we emphasized that our isotope data and the MTT estimates derived from them, serve as a first characterization of the water MTTs in these tropical catchments (P. 13, L. 21–23). Based on the magnitude of the estimated MTTs of stream water (of the order of years), we also recognise that with the available data it is not possible to discern whether these small differences (among catchments MTTs) are actually caused by

differences in land use (P. 12, L. 22–24). We explained the criteria used to assume stationary conditions (Section 2.4 and 3.5), and emphasised the limitation of the MTT estimates due to these assumptions and the sampling period. We included the plots of the best fitting efficiencies and the associated GLUE uncertainties as supplementary material to address uncertainty in the MTT of stream water (Fig. S5–18).

(2) Manuscript structuring

The introduction lacks any logical structure and must be improved. As currently presented, several topics are touch in an apparently random order. First montane landscapes then isotopes then MTT then tracers then EMMA and finally Kenya. The must be a general building of argument to highlight a knowledge gap that this study is trying to fill. The review of literature is

10   rather surficial and must be improved to highlight better the current landscape surround this study to help the reader see where this study fits in with previous work.

Reply: We revised the introduction section following the suggestions of the referee.

In addition, there is some concern with regards to mixing results in with methods. The section 2.5.2 is a good example of this.

15   To alleviate this, I would recommend adding a results section whereby you present the raw data collected (isotopic and chemistry) and characterize these data fully. That type of an overview and statistic presentation will then lay a groundwork for the more advance results. In practice, this means to expand sections 3.1 and 3.2 and allow the data to take center stage for this study – which is valid as these data are a significant contribution to the literature. As such, the data collected should be thoroughly reviewed and presented for the reader.

20   Reply: We expanded Section 3.1 and 3.2 to put more emphasis on the field data collected for this study. We also moved part of the methods section to the results section.

Last, the discussion section left a bit to be desired. I felt there was much text in this section that could find a better home in results as it just highlights the findings of this current study. There could be expansion on the limitations and implications of

25   this study for the region or these types of regions. That shift in emphasis would likely resonate better with readers helping this study move from a place-based investigation to a more general research investigation.

Reply: We followed the suggestions of the referee to put more emphasis on the limitations and implications of this study (Section 4.4). We also moved certain parts to the results section, when we felt that it would be more appropriate.

**Referee #3**

*General comments:*

The mean transit time (MTT) estimates based on a data set covering ca. 1.5 years are likely to be highly uncertain.

Reply: With respect to this concern, we emphasized in the revised manuscript that the presented MTT estimates for stream water, are meant for a first characterization and comparison of the travel times of water between the sub-catchments, assuming stationary conditions. We also emphasized that this information will serve as baseline for future research in which methods like time-variant techniques, could be used: "Due to the low fitting efficiencies of the MTT models, specifically for stream water, we consider the presented MTT estimations as valuable preliminary findings. These can serve as a baseline for future studies, in which more sophisticated methods like time-variant approaches can be used." (P. 13, L. 21–23).

This is evident, for instance, in the similar numbers of NSE, RMSE and Bias for the streamwater and soil water samples at the sites SHA and TF (Table 3): While streamwater was sampled weekly at these sites (n>100), MTT estimates were similarly uncertain for streamwater as for soil water - from which only a small number of samples was collected (n<17)! Thus, based on the model performance criteria presented in the manuscript, I would not strictly believe the values obtained for streamwater either.

Reply: We would like to clarify that the goodness of fit (NSE), RMSE and Bias, are not the same for soil water and stream water. Since the observed isotope signals of soil water have a larger amplitude than of stream water, estimations of MTT of soil waters have, in general, higher NSE, but also higher values of RMSE and Bias. Further, neither the associated uncertainties of estimations of MTT for soil water are comparable to those of stream water: uncertainty ranges for stream water (Table 5) are expressed in years while for soil waters these are in the order of weeks (Table 6). Regarding the number of samples taken for the convolution approach in order to estimate MTTs, for stream water we took $n = 75$ samples for each of the four catchments (not $n > 100$ as suggested by the referee), while for MTT estimations of soil water sites, the number of samples was $n = 46$ (OUT-S50) and $n = 47$ (NF-S15, OUT-S15) (not $n < 17$ as it is stated by the referee).

Although the authors elaborate on the shortcomings of their data set with regard to estimate MTTs (Sect. 4.3), they do not consider using an alternative approach such as the young-water fraction framework (Kirchner, 2016a, b). This framework uses the seasonal cycle amplitudes of streamwater and precipitation amplitudes to estimate the fraction of water younger than ca. 3 months. Thus, with the data set presented by Jacobs et al., such an analysis might result in estimates of the young-water fractions of streamwater that are more robust than the MTTs. (Using the soil water samples from the sites NF and OUT might also reveal some interesting results, however, the data from the sites SHA and TTP are clearly too incomplete for such an analysis.)

Reply: We thank the referee for this suggestion and have included estimates of the Young Water Fractions of stream water (YWF) in our revised manuscript (Section 3.4).

In the catchment SHA, the samples from a wetland (WL, n=4) and the shallow well (WE.b, n=2) comprised two important end-members in the 3-component mixing analysis, whereas no wetlands or shallow wells were sampled in the other two catchments. Thus, I question the comparison made between the three sub-catchments: the relative contribution of precipitation at a site is inevitably linked to the contributions of the other two end members (all components must add up to 1), and therefore the precipitation contributions of NF and TTP cannot simply be compared with the precipitation contributions of SHA.

Reply: The main reason that no shallow wells or wetlands were sampled in the two other sub-catchments, was that there were no similar accessible wetlands in these two sub-catchments. Specifically, the natural forest sub-catchment is highly inaccessible due to the dense vegetation and absence of footpaths. We assume, however, that springs and wetlands represent similar groundwater sources, as supported by their similar chemical composition (P. 11, L. 6–8) and are therefore comparable between the three sub-catchments. This hopefully also clarifies the referee's comment: "[…] wetlands were only analysed for one catchment (SHA), and in the Abstract it appears as if wetlands and springs were considered equivalent end members". With regards to shallow wells, as mentioned in the discussion (P. 13, L. 15–16) there are no shallow wells in the forest, because of absence of habitation, and within the tea plantations shallow wells are not present as all settlements have access to piped water. Therefore, although we tried to include similar end members for each sub-catchment in the design of the study, we were limited in the availability and accessibility of end member sampling sites. We acknowledge in the discussion that the models have considerable uncertainty and that this has led to over- and underestimation of end member contributions, specifically in SHA and TTP (P. 21, L. 7–16).

In general, I find the presentation of the solute concentrations of the different end members and streamwater insufficient - although this data set builds the foundation for the whole study. In the box plot (Figure 2) it is very difficult to distinguish between the different sites (vertical gridlines would help here) and end-members (distinction between the different end members would be impossible in a BW print). I suggest that the authors elaborate more on the data set, incl. uncertainties and times of sampling. Are the times of sampling representative for the flow regime at the sites or were the samples only collected during low-flow conditions? A presentation of the data similar to Figure 4 might be useful for this.

Reply: We changed the presentation of the data from box plots to a table (Table 3), to improve the interpretation of the solute data used in the EMMA. We also added Fig. S1–4, where the time series and sampling times for stream water samples and all end members are indicated, which is similar to Fig. 3 for the isotope data.

Abstract:

- The numbers presented in P. 1, L. 27–29 for the average relative contributions of springs and wetlands to streamwater are confusing: wetlands were only analyzed for one catchment (SHA), and in the Abstract it appears as if wetlands and springs were considered equivalent end members. In addition, I don't understand how the numbers presented in P. 1, L. 29–31 confirm that "... catchment hydrology is strongly influenced by land use, which could have serious consequences for water-related ecosystem services, such as provision of clean water.". Do the authors compare agricultural (i.e., de-forested) catchments to

an un-altered forested catchment (i.e., baseline scenario)? If this is the case, then the results should be presented within such a framework.

Reply: We indeed tried to identify the effect of land use (i.e. forest vs. agriculture) on hydrological flow paths, and in our understanding this is also clear from the aims and hypotheses posed in the introduction: "In this study, we used a combination of mean transit time (MTT) analysis and end member mixing analysis (EMMA) to assess the effect of land use on spatial and temporal dynamics of water sources and flow paths in catchments with contrasting land use (i.e. natural forest, smallholder agriculture and commercial tea and tree plantations) in the Mau Forest Complex." (P. 3, L. 2–5). As explained in a response to an earlier comment of Referee #3, the similarity in solute concentrations in springs and wetlands, we consider that these end members, although named differently, represent the same groundwater source (P. 11, L. 6–8).

Introduction:

The different sub-sections of the introduction should be linked better. For instance, paragraphs 1 and 2 present two very different topics (tropical montane catchments and stable water isotopes, respectively), which have to be put into a common context, otherwise the reader is lost.

Reply: We revised the introduction to address these comments and those of referee 1 and 2.

The authors hypothesize that (a) streamwater in the natural forest sites is (on average) older than streamwater in agricultural catchments (smallholder agriculture, tea and tree plantations); (b) precipitation comprises a larger fraction of streamflow in the agricultural catchments than in the naturally forested catchment; and (c) that seasonality in rainfall causes temporal variability of these streamwater sources throughout the year. The formulations of the working hypotheses (a) and (b) are somewhat redundant: when streamflow at site A contains more precipitation (i.e., "new" water) relative to another site B, we should expect the mean transit time of Site A to be shorter. Thus, hypothesis (a) results from hypothesis (b). Regarding hypothesis (c), I don't understand how accepting/rejecting this statement adds to the conclusions of this study. The authors discuss hypothesis (c) only briefly later in the manuscript (P. 11, L. 21–23), which makes me wonder why it is stated so prominently in the Introduction?

Reply: We agree with the observations of the referee. We revised the hypotheses, merging (a) and (b) and removing (c) due to its lack of relevance for the presented results presented. We formulated a second hypothesis based on recommendations by Referee #1. "Based on these results, we hypothesised that (a) the natural forest sub-catchment has a longer MTT than the tea plantation and the smallholder agriculture sub-catchments, because precipitation contributes less to streamflow in the forest catchment, and (b) the precipitation that contributes directly to streamflow will reach the stream through surface runoff in the tea plantation and smallholder agriculture sub-catchments and through shallow sub-surface flow in the forest sub-catchment." (P. 3, L. 20–24).

Methods:

P. 3, L. 30: What are the areal fractions of different land-use types in the main catchment (OUT)? This information would also be required to elaborate on the authors' statement on P. 13, L. 18–20: "One could also expect that, since OUT is a mixture of the three land use types dominating the sub-catchments, the MTT should be similar to or an average of the estimated MTTs of the sub-catchments.". This statement would only be true if the three sub-catchments are representative for the areal fractions of land use in the main catchment.

We included the areal fractions in the methods: "These were nested in a 1 021 km² large catchment, referred to as the main catchment (OUT), which is characterized by a mixture of these three land use types (NF = 37.6 %, SHA = 51.0 % and TTP = 11.4 %)." (P. 3, L. 29–31).

2.3 Sampling and laboratory analysis: What are the instruments' measurement precision and accuracy? Especially in the case of Li, the measured concentrations (« 1ug/L) might be highly uncertain for precipitation and throughfall.

Reply: We did not have accuracy and precision information for all instruments used, but presented the limits of quantitation in Table S1.

Results:

3.2 Isotopic composition: "There was no significant effect of elevation on 18O values of the precipitation samples, but precipitation samples collected at higher altitude (SHA-PC) were generally more depleted than those collected at lower altitudes (NF-PC, TTP-PC and OUT-PC).". This sentence is confusing, please reformulate.

Reply: We reformulated to: "Precipitation samples collected at higher altitude (SHA-PC) were generally more depleted than those collected at lower altitudes (NF-PC, TTP-PC and OUT-PC), with a change of −0.099 ‰ $\delta^{18}$O per 100 m. However, linear regression analysis revealed there was no effect of elevation on $\delta^{18}$O values of the precipitation samples ($p = 0.08$)." (P. 8, L. 25–28).

Figure 6 and analysis of Figure 6: Some of the relative contributions are highly uncertain, however, I miss a proper uncertainty analysis here. Although the authors discuss various sources of uncertainty in Sect. 4.2., a quantitative uncertainty analysis is still missing. At least, showing the error bars in Figure 6 would be helpful to interpret the results with more caution (i.e., Could the variability of the end members be an artefact of uncertainty in the EMMA?, P. 11 L. 21–23)). In addition, the Abstract, the authors present the average contributions without any uncertainty measures, which might be misleading.

Reply: We ran an uncertainty analysis with bootstrapping and included the uncertainty measures in Fig. 5. We also emphasized the uncertainty in our results in the discussion, to make clear that the results presented in this study are preliminary findings, forming a knowledge base for more in-depth research into the hydrological functioning of these and other African tropical montane catchments.

Discussion:

4.2 Dominant water sources: Based on another study in the NF catchment (Jacobs et al., in review) the authors conclude that in the NF catchment precipitation reaches the stream network via shallow sub-surface flow. Short residence times in the shallow subsurface thus result in dilution effects in streamwater. However, for the TTP catchment, the authors claim that ". . .surface runoff could have a different chemical signature than precipitation. . ." (P. 11, L. 13), which somewhat contradicts their previous statement in L. 3: "Therefore, if event water, i.e. precipitation or throughfall, is only in contact with the soil for a short time (e.g. several hours), the chemical composition of the water that enters the stream might be comparable to the composition of precipitation or throughfall.". Please clarify this.

Reply: Revision of the discussion to address comments by all referees removed this confusing point from the manuscript.

*Specific comments:*

P. 8, L. 13: Were these evaporated samples used in the analysis? Please clarify.

Reply: We added: "Although these samples were not used for the development of the LMWL, they were included in the mean transit time (MTT) analysis." (P. 8, L. 23–24) to clarify.

P. 9, L. 28: ". . . has been observed elsewhere as well." – Where exactly? Are these sites comparable to the sites of this study?

Reply: We changed this sentence to: "This has also been observed in Canada (Ali et al., 2010) and the Brazilian Amazon (Chaves et al., 2008; Germer et al., 2007), and can be attributed to seasonal variations in plant growth and dry and wet atmospheric deposition of K and Mg originating from biomass burning in our study area." (P. 10, L. 28–31).

P. 12, L. 15: An alternative method to sample soil water would be suction lysimeters.

Reply: We agree with the observation of the referee that lysimeters could be used for soil water sampling. We therefore added: "Alternatives for wick samplers, such as suction lysimeters, should be used to avoid contamination of soil water samples." (P. 13, L. 19–20).

P. 14, L. 23: "Due to the similar soils. . ."

Reply: This sentence was removed during revision of the manuscript.

**Referee #4**

*General comments:*

Given that, the available isotopic data is only 1.5 years long the authors should provide an assessment of the uncertainty in the computed MTT? The performance of the fits by the Gamma and EPM are actually similar yet the MTT for OUT_S15 was

5      different between these two models. How do you explain that? It is not clear how the authors chose the Gamma and EPM functions. Did they consider what model had better constrained parameters? In addition to the modelling shortcomings, how can MTT estimates calculated from 1.5 years of data provide information about the hydrologic impacts of different land covers?

Reply: We added a supplement with detailed plots of the best solutions and associated uncertainty ranges for every stream and mobile soil water site (Fig. S5–18). Indeed, the performance of the fits to the objective function (NSE) for Gamma and EPM

10      are quite similar for soil water, but the estimated MTTs are not different. We discuss the performance of both models for soil water in the discussion: "However, a simpler exponential distribution model (EM) might have been equally appropriate, since the parameter range of behaviour solutions of the gamma model (GM) and the exponential piston flow model (EPM) suggest that both models could be simplified to an exponential distribution model (EM). In order to avoid over-parametrization, models with fewer parameters (in this case EM) are preferred when they provide comparable results." (P. 14, L. 2–5). Furthermore,

15      the justification of model selection is described in Section 2.5 (P. 7, L. 13–16): "Two-parameter models such as the gamma model (GM) or the exponential piston flow model (EPM) are commonly used for MTT estimations (Hrachowitz et al., 2010; McGuire and McDonnell, 2006). These models were identified by Timbe et al. (2014) as most suited to infer MTT estimations of spring, stream and mobile soil water in an Andean tropical montane forest catchment, and were therefore applied in our study (Table 2)." We agree that a 1.5-year dataset is not long and that ideally it should cover a period as long as or longer than

20      the MTT, but one has to consider that no data were available for the African montane forest. Furthermore, due to limited funding and accessibility in such remote areas, it is challenging to collect a long-term dataset for stable isotopes. Considering the conditions in the study area and the requirements for MTT analysis, we think it is reasonable to present the estimated MTTs for the three sub-catchments and main catchments as first and preliminary findings, as long as its uncertainty is shown.

25      I wonder if a first step should be a hydrometric analysis that compares land covers and that can informed the findings form the MTT in light of physical processes. In addition, there might be interesting patterns in the isotopic data alone in terms of means per location, per season, comparisons across soil, stream, groundwater, and precipitation that would allow contrasting the different land covers. I am looking a figure 3 thinking: there is many data that have not been appropriately described in the paper. My point is that the isotopic data can we used on other ways different from in convolution equation for MTT.

30      Reply: As suggested by other referees, we expanded the presentation of the raw isotope data in Section 3.2 to give it a more prominent position in the manuscript. This hopefully also addresses the concern of Referee #4 that not all data has been described appropriately in the paper. Additionally, we included the calculation and analysis of the young water fraction (YWF) (Kirchner, 2016) of the analysed catchments (Section 3.4).

The organization of the paper and its content is insufficient. a. The introduction is no short and does not set up the problem well. It is not clear what would the contribution of this study be nor how it fits with previous literature.

Reply: The introduction section is completely revised based on suggestions by several referees.

5 b. Methods: It to short and refers the reader to a paper in review. A more comprehensive description is in order. The methods indicated that precipitation was estimated using Thiessen polygons based on the information (I assumed, from the nine tipping buckets) however the results from this analysis is never presented in the results section. How variable is precipitation in space and time in this system?

Reply: The current version of the manuscript is already quite long. The study area and collection of discharge and precipitation
10 has been described extensively in other publications (e.g. Jacobs et al. 2017 and the manuscript under review, which is now published as Jacobs et al. 2018). We therefore decided not to repeat this in the current manuscript. The precipitation results have indeed not been presented separately in the results, as this has been done elsewhere and is not the objective of this manuscript, but rainfall data is displayed in Fig. 4 as weekly precipitation. This also clearly shows the temporal variation and differences in precipitation between the four catchments.

c. The result section is vague. For instance on 3.1. (Solute concentrations) the authors do not describe any one solute but instead talk all simultaneously as high or low. The result sections should include some actual numbers so that the reader knows what low or high mean. Likewise, there is no information in the results about how the values for the isotopic concertation vary in space and time per precipitation, stream, soil water, etc.

20 Reply: The challenge with the presentation of a large dataset, as used for this analysis, is that the text can become very long when all solutes for all sites and end members are described in detail. We modified Section 3.1 to include more detailed information, but decided against explaining all variations in solute concentrations in detail in the text, as this would make the results section unnecessarily long. We nevertheless hope that the modifications provide a satisfactory level of detail to address the concern of Referee #4. With regards to the variations in isotopic values, we added a table with mean, standard deviation,
25 minimum and maximum values for $\delta^{18}O$ to the manuscript (Table 4).

**References**

[revised manuscript text omitted]

---

## Referee Report (RR1)

Comment to "Assessment of hydrological pathways in East African montane catchments under different land use" by Suzanne R. Jacobs et al.

General comments:

I acknowledge the effort made by the authors to follow the intricate suggestions given during the first revision. Nevertheless, there are some aspects relevant to the determination of MTTs in the stream waters that yet deserve a particular attention.

There are some misunderstandings on the issues about the MTT determination using water stable isotopes in the new manuscript; these issues may be abridged as it follows:

1) The use of any seasonal variation of tracer input signal for the determination of MTTs, using any type of model, is limited to a short range of years because the characterisation of the tracer signal is limited by the occurrence of spurious errors such as the analytical ones. DeWalle et al. (1997) showed that 5 years is the realistic limit for an exponential TTD.

2) It has been shown (e.g. Stewart et al., 2010; 2012) that MTT determinations using any seasonally varying input signal typically underestimate the old tails of the TTDs and subsequently the corresponding MTTs (because the relationship between the signal modification and TT is very non-linear; Kirchner, 2016a ).

3) A frequent case of 2) is produced when young waters are mixed with old ones; the form of the TTD may become very dissimilar and the MTT of the mixed water becomes strongly underestimated (Kirchner, 2016a).

The authors cannot therefore suggest that applying "more sophisticated methods like time-variant approaches" using the same stable water isotopes data may help to improve their MTT results. Looking for a better determination of MTTs, when they are of the order of several years, is not possible with these data but using other tracers, as sensibly suggested by the authors. Instead, as far as I know, the more advanced and reliable approach using stable water isotopes for analysing catchment waters ages is the unsophisticated analysis of the young water fraction (Fyw) for different stream discharge ranges as proposed by Kirchner (2016b) and implemented by von Freyberg et al. (2018). I would not recommend the use of tracer signal standard deviation instead of sinusoid amplitude because spurious errors may be important for much damped signals as well as for precipitation input signal.

This approach might be very adequate to the purpose of the authors, because different Fyw sensitivities to discharge might be identified in the diverse sub-basins, demonstrating different behaviours of the runoff generation processes.

Detailed comments:

Page 2, lines 1-3: As commented before, this is not a model issue but a tracer one.

Page 6, lines 22-33: Steady state conditions refer primarily to time and homogeneity refers primarily to space; If one property is given, this do not necessarily imply that the other is also given.

In the case of the studied basins, none of both assumptions (homogeneity or stationarity) may be sensibly claimed, given both their large sizes and the diverse water sources analysed in the paper. The relatively similar Fyw assessed for the catchment waters, obtained following a time-weighted approach, may hide their dependences on discharge and possible differences of these dependences among catchments (von Freyberg et al, 2018).

Page 7, lines 23-25: As already stated in the first review, using as behavioural only the parameter sets that are 5% below the best efficiency is not adequate when the best efficiency is so low and results in artificially reduced uncertainties. For instance, on table 5, the GM model applied to TTP-RV yielded a MTT of 3.3 (2.8-4.3) with a NSE of 0.05; for n=75, this NSE has a probability of the null hypothesis higher than 0.05: although all the parameter sets should be discarded as non-behavioural, a short uncertainty range is claimed.

Page 9, lines 27-29: discussed above.

Page 10, lines 11-19: This is a severe argument against table 5, so this table should be changed.

Page 13, lines 22-27: As discussed before, the problem is the tracer, not the model.

Page 13, lines 30-33: The research should be focused to Fyw and its dependence on discharge, instead on MTT.

References:

DeWalle DR, Edwards PJ, Swistock BR, Aravena R, Drimmie RJ. (1997). Seasonal isotope hydrology of three Appalachian forest catchments. Hydrological Processes 11(15): 1895–1906.

Kirchner, J. W. (2016a). Aggregation in environmental systems-Part 1: Seasonal tracer cycles quantify young water fractions, but not mean transit times, in spatially heterogeneous catchments, Hydrol. Earth Syst. Sci., 20, 279-297.

Kirchner, J. W.(2016b). Aggregation in environmental systems-Part 2: Catchment mean transit times and young water fractions under hydrologic nonstationarity, Hydrol. Earth Syst. Sci., 20, 299-328.

Stewart MK, Morgenstern U, McDonnell JJ. (2010). Truncation of stream residence time: how the use of stable isotopes has skewed our concept of streamwater age and origin. Hydrological Processes 24: 1646–1659.

Stewart MK, Morgenstern U, McDonnell JJ, Pfister L. (2012). The 'hidden streamflow' challenge in catchment hydrology: a call to action for stream water transit time analysis. Hydrological Processes 26: 2061–2066.

von Freyberg, J., Allen, S. T., Seeger, S., Weiler, M., and Kirchner, J. W. (2018): Sensitivity of young water fractions to hydro-climatic forcing and landscape properties across 22 Swiss catchments, Hydrol. Earth Syst. Sci., 22, 3841-3861.

---

## Author Response (AR2)

Dear Editor, dear Referees,

We would like to thank you for the positive response and helpful feedback. We addressed the comments of all referees, as detailed below in blue. The page and line numbers refer to the currently revised manuscript (no tracked changes). More specifically, we improved our estimation of the young water fraction based on the methods outlined by Kirchner (2016a). However, we limited the analysis of the discharge-dependence of the young water fraction to two discharge classes, due to the size of the dataset and the uncertainty associated with small sample sizes. We further highlighted the limitations of the used MTT estimations and value of the more robust young water fraction estimation. With regard to the confidence bounds, we removed the modelled parameters and corresponding confidence bounds for sites with low NSE from Table 5.

We hope we have addressed any shortcomings, and look forward to hearing your decision on the suitability of the manuscript for publication.

Kind regards,

Suzanne Jacobs

**Referee #1**

General comments:

I acknowledge the effort made by the authors to follow the intricate suggestions given during the first revision. Nevertheless, there are some aspects relevant to the determination of MTTs in the stream waters that yet deserve a particular attention.

There are some misunderstandings on the issues about the MTT determination using water stable isotopes in the new manuscript; these issues may be abridged as it follows:

1) The use of any seasonal variation of tracer input signal for the determination of MTTs, using any type of model, is limited to a short range of years because the characterisation of the tracer signal is limited by the occurrence of spurious errors such as the analytical ones. DeWalle et al. (1997) showed that 5 years is the realistic limit for an exponential TTD.

2) It has been shown (e.g. Stewart et al., 2010; 2012) that MTT determinations using any seasonally varying input signal typically underestimate the old tails of the TTDs and subsequently the corresponding MTTs (because the relationship between the signal modification and TT is very non-linear; Kirchner, 2016a).

3) A frequent case of 2) is produced when young waters are mixed with old ones; the form of the TTD may become very dissimilar and the MTT of the mixed water becomes strongly underestimated (Kirchner, 2016a).

The authors cannot therefore suggest that applying "more sophisticated methods like timevariant approaches" using the same stable water isotopes data may help to improve their MTT results. Looking for a better determination of MTTs, when they are of the order of several years, is not possible with these data but using other tracers, as sensibly suggested by the authors. Instead, as far as I know, the more advanced and reliable approach using stable water isotopes for analysing catchment waters

ages is the unsophisticated analysis of the young water fraction (Fyw) for different stream discharge ranges as proposed by Kirchner (2016b) and implemented by von Freyberg et al. (2018). I would not recommend the use of tracer signal standard deviation instead of sinusoid amplitude because spurious errors may be important for much damped signals as well as for precipitation input signal. This approach might be very adequate to the purpose of the authors, because different Fyw

5   sensitivities to discharge might be identified in the diverse sub-basins, demonstrating different behaviours of the runoff generation processes.

Reply: We thank the referee for the advise on the use of the young water fraction ($F_{yw}$) to improve our understanding of runoff generation processes in our study area. As recommended, we calculated the sinusoid amplitude of the tracer signals and used this instead of standard deviation to estimate the $F_{yw}$. Additionally, we estimated the $F_{yw}$ for samples during low and high

10   discharge, which corresponds to the classification used to differentiate between high and low flow end member contributions and in the conceptual model (Fig. 5–6). Due to the sample size, classification into more discharge classes with a lower number of samples per class would result in higher uncertainty, as highlighted by Von Freyberg et al. (2018). We also acknowledge the limitations of the use of $\delta^{18}O$ and $\delta^2H$ to estimate MTT in catchments with long transit times (e.g. P. 14, L. 4–8) and therefore removed the reference to using more sophisticated approaches using the same tracers as a way to improve the MTT

15   estimations.

Detailed comments:

P. 2, L. 1–3: As commented before, this is not a model issue but a tracer one.

Reply: Based on the recommendation by the referee, we changed the sentence to: "The results further suggest that the selected

20   transit time models and tracers might not be appropriate in tropical catchments with highly damped stream water isotope signatures. A more in-depth investigation of the discharge-dependence of the young water fraction and transit time estimation using other tracers, such at tritium, could therefore shed more light on potential land use effects on the hydrological behaviour of tropical montane catchments." (P. 2, L. 2–6).

25   P. 6, L. 22–33: Steady state conditions refer primarily to time and homogeneity refers primarily to space; If one property is given, this do not necessarily imply that the other is also given. In the case of the studied basins, none of both assumptions (homogeneity or stationarity) may be sensibly claimed, given both their large sizes and the diverse water sources analysed in the paper. The relatively similar Fyw assessed for the catchment waters, obtained following a time-weighted approach, may hide their dependences on discharge and possible differences of these dependences among catchments (von Freyberg et al,

30   2018).

Reply: We removed the statement on the homogeneity and steady state in the study area, based on the similarity in $F_{yw}$ between the sub-catchments. We added the $F_{yw}$ analysis as an additional method to complement the other analyses. We now highlight the potential inappropriateness of assuming homogeneity and steady state: "According to Kirchner (2016a), the estimation of MTT through tracer cycles and methods like the lumped convolution approach should be limited to homogeneous catchments

for which steady state conditions apply. Because we cannot be certain of the degree of homogeneity and steady state in our study area, we complemented the analysis with the more robust calculation of the young water fraction $F_{yw}$ (Kirchner, 2016a)." (P. 7, L. 20–23).

5   P. 7, L. 23–25: As already stated in the first review, using as behavioural only the parameter sets that are 5% below the best efficiency is not adequate when the best efficiency is so low and results in artificially reduced uncertainties. For instance, on table 5, the GM model applied to TTP-RV yielded a MTT of 3.3 (2.8-4.3) with a NSE of 0.05; for n=75, this NSE has a probability of the null hypothesis higher than 0.05: although all the parameter sets should be discarded as non-behavioural, a short uncertainty range is claimed.

10   Reply: We removed the estimated parameters for sites and models with a low NSE, i.e. TTP-RV (gamma model) and all stream water sites (exponential piston flow model), from Table 5, and therefore also the claimed short uncertainty range.

P. 9, L. 27–29: discussed above.

Reply: As mentioned in our earlier reply, we removed our statement on homogeneity of our study area based on similarity in

15   $F_{yw}$ between the three sub-catchments.

P. 10, L. 11–19: This is a severe argument against table 5, so this table should be changed.

Reply: We changed the table as detailed in our previous response.

20   P. 13, L. 22–27: As discussed before, the problem is the tracer, not the model.

Reply: Following the earlier comment of the referee, we removed the paragraph from the manuscript and remain with the use of tritium or other tracers as more appropriate methods to estimate MTT (P. 14, L. 4–8)

P. 13, L. 30–33: The research should be focused to Fyw and its dependence on discharge, instead on MTT.

25   Reply: We thank the referee for this suggestion. The inclusion of the $F_{yw}$ is definitely a valuable addition to the manuscript. However, we think that an in-depth analysis of the dependence of the $F_{yw}$ on discharge requires a larger dataset, as a low number of samples per discharge class would lead to considerable uncertainty. Therefore, we would like to keep the MTT analysis, now backed by an investigation of $F_{yw}$.

25   estimated MTTs and the low fraction of young water (YWF) suggest that the majority stream water in all catchments originates from 'old' water or groundwater.". At the end of this paragraph, the authors point out that (P. 12, L. 30–31): "The importance of groundwater does, however, contradict the generally high contribution of precipitation and throughfall to streamflow in most catchments.". However, an explanation for these opposite results is missing here. Instead, the section continues to discuss the MTT's of soil water. Only later in Sect. 4.4 the authors point towards the large uncertainties in the EMMA, as well as to
30   the uncertainties in the transit time models, without clearly indicating which result of their two analysis methods is (if at all) the most reliable. As a consequence, the reader might be left puzzled about the validity of the presented results as well as the conclusions drawn from them (i.e., flow processes). I would therefore encourage the authors to discuss the opposite results for NF and SHA more thoroughly.

Reply: We addressed the validity of the results by including "Due to the weaknesses associated with both methods, there is considerable uncertainty in the estimated MTT and end member contributions. However, supported by the young water fraction analysis, we were able to draw a reliable conclusion about the importance of groundwater during low and high flows in the different land use types." (P. 15, L. 4–7) in the conclusion.

5    We moved the sentence about the discrepancy between the high precipitation contribution and the importance of groundwater in NF and SHA to Sect. 4.3 (P. 12, L. 17), where we explain that the high precipitation contribution in SHA is most likely an overestimation due to the selection of SHA-WE.b as end member.

*Specific comments:*

10   P. 11, L. 28–30: "The end member mixing analysis (EMMA) showed that precipitation (PC) was one of the three selected end members in all catchments, as depicted in our conceptual model of the rainfall–runoff generation processes in the three sub-catchments with different land use (Fig. 6)." This is not logical: you select PC as an end-member in the EMMA and then present this selection of PC a s a result? Do you mean instead that "precipitation (PC) was the dominant contribution in streamflow in all catchments"?

15   Reply: We thank the referee for pointing out this confusing sentence. As mentioned in Sect. 2.4, the three end members for each sub-catchment were not selected *a-priori*, but in the processes of assessing how many end members were required and which of the sampled end members fit best to the stream water samples. In that sense, the selection of PC is a result of the analysis. However, the referee is correct that the actual EMMA can only be carried out after selection. We therefore changed the sentence to: "The end member mixing analysis (EMMA) showed that precipitation (PC) was an important end member in

20   all catchments, as depicted in our conceptual model of the rainfall–runoff generation processes in the three sub-catchments with different land use (Fig. 6)." (P. 12, L. 4–6).

P. 11, L. 32: Include "high" before "hydraulic conductivity"

Reply: Thank you, we corrected this in the manuscript (P. 12, L. 8).

P. 12, L. 15: "The latter…" Use "This" instead, since only one end-member is mentioned in the previous sentence.

Reply: We changed this in the manuscript (P. 12, L. 19).

[revised manuscript text omitted]

---

## Author Response (AR3)

Dear Editor,

We addressed the final corrections and hope that the manuscript now meets the requirements. Please find a point by point explanation below in blue.

On behalf of all authors, I would like to thank you for your support throughout the review process.

Kind regards,
Suzanne Jacobs

Dear authors,

thank you very much for addressing the remaining reviewer comments. I think you have done this in a meaningful way that helps the reader to better understand the limitations of the presented analysis.

Before I can accept your manuscript for publication, I would like to ask you to do a few small technical edits:

(1) please rearrange table 5 (and 6) in a way that avoids giving the same information twice (e.g. catchment areas,...) and to add the estimates of young water fraction for each catchment
We rearranged the rows in Table 5 and 6 in such a way that information is only given once per site. We added a column to Table 5 with the estimated young water fractions.

(2) in figure 3, please make sure to use consistent units: precipitation is flux in the system and needs to be expressed as a rate (volume/time, i.e. in your case as mm/week). Please also discharge to mm/d.
We corrected the unit for precipitation to mm/week and recalculated specific discharge to mm/d.

(2) in figure 5 it is difficult to distinguish the endmember contributions as some colors are too similar. please find a clearer color coding. also, i cannot see the blue end member (shallow well (WE.a) )
We changed the colour coding of the different end members. We also removed the end members that were not included in the final EMMA models of the catchments from the legend of Figure 5.

(3) please make it clearer in the methods section that the EMMA is based on Principal Component analysis
We added a sentence to Section 2.4 to clarify that principle component analysis is used for the EMMA: 
[revised manuscript text omitted]

| Site[a] | n[b] | Elevation | Observed $\delta^{18}O$ | | Model[d] | Modelled $\delta^{18}O$ | | | | | | |
|---|---|---|---|---|---|---|---|---|---|---|---|---|
| | | | Mean | SD[c] | | Mean | SD[c] | NSE[d] NSE[e] | RMSE[e] RMSE[f] | Bias | MTT[f] MTT[g] | $\alpha/\eta$[g] $\eta$[h] |
| | - | m | ‰ | ‰ | | ‰ | ‰ | - | ‰ | ‰ | weeks | - |
| *Gamma model (GM)* | | | | | | | | | | | | |
| NF-S15 | 47 | 1 971 | −1.62 | 1.64 | GM | −1.74 | 1.48 | 0.79 | 0.75 | −0.12 | 3.2 (2.8–4.1) | 1.5 (0.9–2.2) |
| | | | | | EPM | −1.67 | 1.38 | 0.78 | 0.77 | −0.05 | 3.3 (2.6–4.4) | 1.0 (0.9–1.1) |
| OUT-S15 | 47 | 1 721 | −0.68 | 1.20 | GM | −0.71 | 0.99 | 0.50 | 0.84 | −0.03 | 7.9 (6.1–11.3) | 0.9 (0.6–1.2) |
| | | | | | EPM | −0.58 | 0.94 | 0.52 | 0.82 | 0.11 | 4.5 (3.2–6.7) | 0.8 (0.7–1.1) |
| OUT-S50 | 46 | 1 721 | −0.84 | 1.35 | GM | −0.92 | 0.93 | 0.47 | 0.97 | −0.08 | 10.4 (8.8–12.6) | 1.4 (1.1–2.0) |
| | | | | | EPM | −0.90 | 0.85 | 0.46 | 0.99 | −0.06 | 10.8 (8.0–13.9) | 1.0 (0.9–1.3) |
| *Exponential piston flow model (EPM)* | | | | | | | | | | | | |
|  |  |  |  |  | |  |  |  |  |  |  |  |
|  |  |  |  |  | |  |  |  |  |  |  |  |
|  |  |  |  |  | |  |  |  |  |  |  |  |

[a] NF = natural forest, OUT = main catchment, S15 = mobile soil water 15 cm depth, S50 = mobile soil water 50 cm depth; [f] number of samples; [c] standard deviation; [d] GM = gamma model, EPM = exponential piston flow model; [d,e] Nash-Sutcliffe efficiency of objective function; [e,f] root mean square error; [f,g] predicted mean transit time (in weeks); [g,h] model parameters for GM ($\alpha$) and EPM ($\eta$).